# Absolute quantification of cohesin, CTCF and their regulators in human cells

Johann Holzmann[1,2,3†‡], Antonio Z Politi[4†§], Kota Nagasaka[1], Merle Hantsche-Grininger[4], Nike Walther[4#], Birgit Koch[4¶], Johannes Fuchs[1,2,3**], Gerhard Dürnberger[1,2,3], Wen Tang[1], Rene Ladurner[1††], Roman R Stocsits[1], Georg A Busslinger[1], Béla Novák[5], Karl Mechtler[1,2,3], Iain Finley Davidson[1*], Jan Ellenberg[4*], Jan-Michael Peters[1,6*]

[1]Research Institute of Molecular Pathology (IMP), Vienna Biocenter (VBC), Vienna, Austria; [2]Institute of Molecular Biotechnology of the Austrian Academy of Sciences (IMBA), Vienna Biocenter (VBC), Vienna, Austria; [3]Gregor Mendel Institute, Austrian Academy of Sciences, Vienna Biocenter (VBC), Vienna, Austria; [4]Cell Biology and Biophysics Unit, European Molecular Biology Laboratory (EMBL), Heidelberg, Germany; [5]Department of Biochemistry, University of Oxford, Oxford, United Kingdom; [6]Medical University of Vienna, Vienna, Austria

*For correspondence:
davidson@imp.ac.at (IFD);
jan.ellenberg@embl.de (JE);
Jan-Michael.Peters@imp.ac.at (J-MP)

[†]These authors contributed equally to this work

Present address: [‡]Technical Research and Development Novartis, Kundl, Austria; [§]Max Planck Institute for Biophysical Chemistry, Goettingen, Germany; [#]Department of Molecular and Cell Biology, Li Ka Shing Center for Biomedical and Health Sciences, CIRM Center of Excellence, University of California, Berkeley, Berkeley, United States; [¶]Proteomics Core Facility, University of Gothenburg, Gothenburg, Sweden; [**]Department of Biochemistry, Stanford University, Stanford, United States; [††]Hubrecht Institute, Royal Netherlands Academy of Arts and Sciences, University Medical Center Utrecht, Utrecht, Netherlands

**Abstract** The organisation of mammalian genomes into loops and topologically associating domains (TADs) contributes to chromatin structure, gene expression and recombination. TADs and many loops are formed by cohesin and positioned by CTCF. In proliferating cells, cohesin also mediates sister chromatid cohesion, which is essential for chromosome segregation. Current models of chromatin folding and cohesion are based on assumptions of how many cohesin and CTCF molecules organise the genome. Here we have measured absolute copy numbers and dynamics of cohesin, CTCF, NIPBL, WAPL and sororin by mass spectrometry, fluorescence-correlation spectroscopy and fluorescence recovery after photobleaching in HeLa cells. In G1-phase, there are ~250,000 nuclear cohesin complexes, of which ~ 160,000 are chromatin-bound. Comparison with chromatin immunoprecipitation-sequencing data implies that some genomic cohesin and CTCF enrichment sites are unoccupied in single cells at any one time. We discuss the implications of these findings for how cohesin can contribute to genome organisation and cohesion.

DOI: https://doi.org/10.7554/eLife.46269.001

## Introduction

In interphase, eukaryotic genomes form long-range interactions that lead to the formation of chromatin loops and topologically associating domains (TADs) (*Dixon et al., 2012*; *Nora et al., 2012*; *Rao et al., 2014*). These interactions organise chromatin structurally and contribute to gene regulation and recombination. Cohesin complexes are not only required for the formation of chromatin loops and TADs (*Gassler et al., 2017*; *Rao et al., 2017*; *Schwarzer et al., 2017*; *Wutz et al., 2017*), but also for sister chromatid cohesion, a prerequisite for chromosome segregation in mitosis and meiosis (reviewed in *Morales and Losada, 2018*). In interphase, cohesin is enriched at specific positions genome-wide, most of which are also associated with CCCTC-binding factor (CTCF) (*Parelho et al., 2008*; *Wendt et al., 2008*). CTCF consensus binding sites are frequently oriented convergently at TAD borders (*Rao et al., 2014*; *de Wit et al., 2015*; *Guo et al., 2015*; *Vietri Rudan et al., 2015*) and depletion of CTCF leads to a reduction in insulation between TADs (*Nora et al., 2017*; *Wutz et al., 2017*). The mechanism by which cohesin and CTCF contribute to the generation of TADs is unknown, but it has been proposed that cohesin acts by extruding loops of DNA until it

encounters convergently-oriented CTCF sites (*Sanborn et al., 2015*; *Fudenberg et al., 2016*), thus generating long-range interactions by tethering distal regions of the same chromosome together. Additionally, cohesin is thought to mediate sister chromatid cohesion by physically entrapping two different DNA molecules, one from each sister chromatid (*Gruber et al., 2003*; *Ivanov and Nasmyth, 2005*; *Haering et al., 2008*). How cohesin can perform these two apparently distinct functions is poorly understood.

Cohesin is a ring-shaped protein complex composed of four core subunits. The subunits SMC1, SMC3 and SCC1 (also called Rad21 or Mcd1) form a tripartite ring structure and associate via SCC1 with a fourth subunit, which exists in two isoforms in mammalian somatic cells termed STAG1 and STAG2 (also known as SA1 and SA2). Cohesin's binding to and release from chromosomes are mediated by the proteins NIPBL and WAPL, respectively. Recruitment of cohesin to chromatin in vivo depends on NIPBL and its binding partner MAU2 (*Ciosk et al., 2000*; *Gillespie and Hirano, 2004*; *Takahashi et al., 2004*; *Watrin et al., 2006*; *Schwarzer et al., 2017*). NIPBL also stimulates cohesin's ATPase activity in vitro, an activity thought to be essential for loading of cohesin onto DNA (*Arumugam et al., 2003*; *Weitzer et al., 2003*; *Hu et al., 2011*; *Ladurner et al., 2014*; *Murayama and Uhlmann, 2014*; *Petela et al., 2018*). WAPL, on the other hand, is required for cohesin's release from chromatin in interphase and prophase, presumably via opening of the cohesin ring (*Gandhi et al., 2006*; *Kueng et al., 2006*; *Chan et al., 2012*; *Buheitel and Stemmann, 2013*; *Eichinger et al., 2013*; *Tedeschi et al., 2013*; *Huis in 't Veld et al., 2014*). Upon experimental depletion of WAPL in interphase cells, cohesin relocalises to axial elements termed vermicelli (*Tedeschi et al., 2013*). This coincides with global compaction of chromatin that is detectable via DNA and chromatin staining and also by mapping long-range chromatin interactions via Hi-C, indicating that cohesin turnover on chromatin is essential for normal genome organisation (*Tedeschi et al., 2013*; *Gassler et al., 2017*; *Haarhuis et al., 2017*; *Wutz et al., 2017*). In S and G2 phase of the cell cycle, those cohesin complexes that mediate cohesion are protected from WAPL's releasing activity by the protein sororin, which is essential for maintaining sister chromatid cohesion (*Rankin et al., 2005*; *Schmitz et al., 2007*) in the presence of WAPL (*Nishiyama et al., 2010*; *Ladurner et al., 2016*).

Although the regulation of cohesin – chromatin interactions has been well studied, quantitative insight into how cohesin contributes to chromatin loop and TAD formation is lacking. Most Hi-C studies into cohesin-mediated chromosome organisation are performed on a population of cells. The few studies that used Hi-C to investigate the genome organisation of single cells have found that chromosome organisation is variable from cell to cell (*Nagano et al., 2013*; *Flyamer et al., 2017*; *Gassler et al., 2017*; *Nagano et al., 2017*; *Stevens et al., 2017*), raising the possibility that TADs might be the product of ongoing loop extrusion events that occur stochastically and are detectable only when averaging across a cell population. Recent microscopy studies have reported structural features consistent with a TAD-like organisation in single cells (*Boettiger et al., 2016*; *Bintu et al., 2018*; *Szabo et al., 2018*), however cell to cell heterogeneity was also detected (*Bintu et al., 2018*).

To gain insight into how cohesin might function within a single cell, we have used quantitative mass spectrometry (MS) (*Picotti and Aebersold, 2012*) to determine the copy number of soluble and chromatin-bound cohesin complexes and ChIP-seq to analyze cohesin's genomic distribution in populations of HeLa cells synchronised in G1, G2 and prometaphase. We combined these ensemble approaches with automated fluorescence-correlation spectroscopy (FCS) and fluorescence recovery after photobleaching (FRAP) to determine cohesin copy number and residence time on chromatin in individual synchronised HeLa cells. Our findings, as well as those reported in mouse embryonic stem cells and the human cell line U2OS (*Cattoglio et al., 2019*), suggest that a fraction of cohesin and CTCF enrichment sites along chromosome arms may be unoccupied in a single cell at any one time. We discuss the implications of these findings for how cohesin might contribute to genome organisation and sister chromatid cohesion.

## Results

### Mass spectrometry analysis of cohesin copy number

To determine the number of cohesin complexes that exist in HeLa cells, a widely used human cell line, we synchronised cells in G1 phase, G2 phase or prometaphase using thymidine and nocodazole arrest/release protocols. We verified synchronisation efficiency using fluorescence activated cell sorting (FACS) of propidium iodide-stained cells (*Figure 1—figure supplement 1A*) and determined the number of cells collected in each condition. We separated soluble proteins from chromatin-bound proteins by differential centrifugation and released proteins from chromatin by DNase and RNase treatment. Soluble proteins were isolated from between 1400 and 4,200 cells, depending on cell cycle stage, and chromatin-bound proteins were isolated from 62,500 ± 3,100 cells. Liquid chromatography-MS (LC-MS) analyses using an LTQ Orbitrap Velos revealed that each fraction was enriched in marker proteins known to be soluble (glucose metabolising enzymes) and chromatin-bound (core histones), respectively (*Figure 1—figure supplement 1B*). For absolute quantification, an aliquot of each sample was combined with an equimolar mixture of isotopically labelled proteotypic reference peptides generated with the equimolarity through equaliser peptide (EtEP) strategy (*Holzmann et al., 2009*). This reference set consisted of one peptide from SMC1, five peptides from SMC3, three peptides from SCC1 and STAG1 and two from STAG2 (*Appendix 1—table 1*). Scheduled selective reaction monitoring (SRM) analyses of the samples obtained in two experiments were each performed in technical duplicates on a 5500 QTRAP instrument.

Using this approach, we found that the cohesin subunits SMC3 and SCC1 were present in approximately 417,000 and 350,000 copies per G1 cell (*Figure 1*, see *Table 1* for exact values and *Figure 1—figure supplement 2* for individual peptide counts). The excess SMC3 detected over SCC1 is consistent with the previously reported existence of SMC1-SMC3 dimers not bound to SCC1 (*Losada et al., 2000*; *Sumara et al., 2000*; *Waizenegger et al., 2000*). However, consistent with a 1:1:1 stoichiometry of these subunits on chromatin, between 61,000 and 69,000 copies of SMC1, SMC3 and SCC1 were detectable on chromatin in G1. STAG2 has been reported to be in excess of STAG1 in HeLa cells (3:1 in HeLa nuclear extract, *Losada et al., 2000*; ~12–15:1 in SCC1 immunoprecipitates from HeLa total cell extract, *Holzmann et al., 2011*). We consistently detected more STAG2 than STAG1 in all conditions (*Figure 1*, *Table 1*). We detected around 15,000 copies of chromatin-bound STAG1 and 45,000 copies of STAG2 in G1, suggesting that cohesin-STAG2 is present in approximately three-fold excess over cohesin-STAG1 on chromatin in HeLa cells in G1; this ratio increased to 4.6-fold in G2 and decreased to 2.9-fold in prometaphase. The combined total of STAG1 and STAG2 on chromatin was 60,000; taken together this suggests that, on average,

**Table 1.** LC-MS quantification of cohesin subunit copy number.

Absolute quantification of cohesin subunits in chromatin or soluble extracts isolated from G1, G2 or prometaphase synchronised HeLa cells, adjusted for cell number. Data are tabulated as [mean - s.d., mean + s.d.] of two biological replicates and two technical replicates (s.d. = standard deviation). Values in bold or italics derive from quantification of one peptide or two peptides, respectively. Underlined values derive from quantification of a single biological replicate. For individual peptide counts, see *Figure 1—figure supplement 2*.

| Protein | G1 | | G2 | | Prometaphase | |
| | Chromatin-bound | Soluble | Chromatin-bound | Soluble | Chromatin-bound | Soluble |
|---|---|---|---|---|---|---|
| SMC1 | **61125 [47609, 74640]** | - | **142290 [120274, 164306]** | - | <u>**12461 [11497, 13425]**</u> | - |
| SMC3 | 66430 [53421, 79440] | 350228 [312360, 388095] | 158478 [129846, 187111] | 622117 [557712, 686522] | 14165 [12173, 16158] | 899510 [816643, 982376] |
| SCC1 | 69275 [54577, 83972] | 281140 [199837, 362444] | 149500 [127373, 171627] | 356850 [270483, 443217] | 13973 [10478, 17468] | 494155 [314240, 674070] |
| STAG1 | 14718 [11295, 18140] | <u>114355 [104557, 124152]</u> | 23017 [17947, 28087] | <u>120960 [119860, 122061]</u> | 2797 [1842, 3752] | <u>152033 [147488, 156577]</u> |
| STAG2 | *44812 [37745, 51879]* | <u>144160 [131233, 157086]</u> | *106106 [93591, 118621]* | 235980 [213879, 258081] | 8112 [6863, 9361] | 336062 [305203, 366922] |

DOI: https://doi.org/10.7554/eLife.46269.008

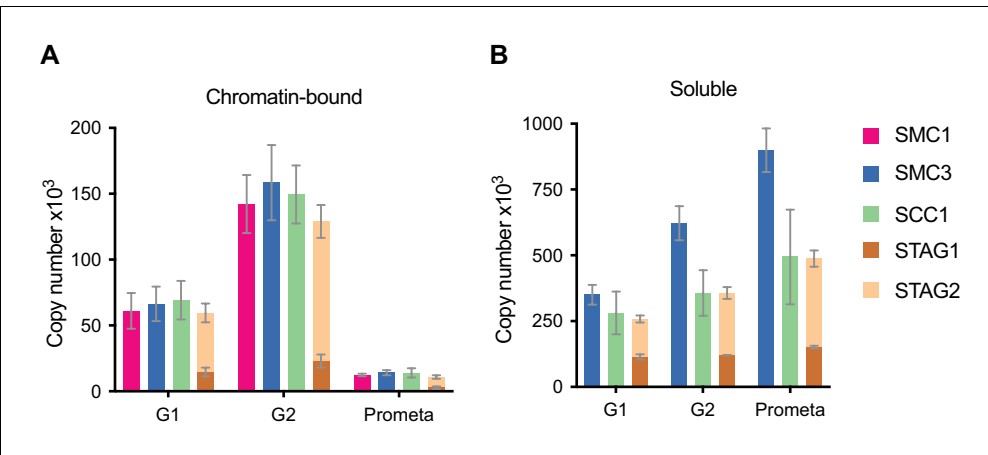

**Figure 1.** LC-MS quantification of cohesin subunit copy number. Absolute quantification of cohesin subunits in (**A**) chromatin or (**B**) soluble extracts isolated from G1, G2 or prometaphase synchronised HeLa cells, adjusted for cell numbers to derive copy number per cell. Data are plotted as mean ± standard deviation from two biological replicates and two technical replicates. For absolute values and exceptions, see *Table 1*. For individual peptide counts, see *Figure 1—figure supplement 2*.

DOI: https://doi.org/10.7554/eLife.46269.002

The following source data and figure supplements are available for figure 1:

**Source data 1.** The Microsoft Excel file lists all proteins identified by LC-MS in the SCC1 immunoprecipitations used to determine cohesin stoichiometry (*Figure 1—figure supplements 3* and *4*).

DOI: https://doi.org/10.7554/eLife.46269.007

**Figure supplement 1.** LC-MS cell synchronisation and fractionation efficiency.

DOI: https://doi.org/10.7554/eLife.46269.003

**Figure supplement 2.** LC-MS quantification of cohesin subunit copy number.

DOI: https://doi.org/10.7554/eLife.46269.004

**Figure supplement 3.** LC-MS quantification of cohesin subunit stoichiometry.

DOI: https://doi.org/10.7554/eLife.46269.005

**Figure supplement 4.** LC-MS quantification of cohesin subunit stoichiometry.

DOI: https://doi.org/10.7554/eLife.46269.006

between 60,000 and 70,000 cohesin complexes exist in chromatin fractions isolated by differential centrifugation from HeLa cells in G1 phase.

## The stoichiometry of cohesin complexes remains constant throughout G1, G2 and prometaphase

As described above, cohesin functions both in genome organisation and in sister chromatid cohesion. Models for cohesin function during these processes have proposed that cohesin might exist in a variety of stoichiometries (*Nasmyth, 2011*; *Hassler et al., 2018*). According to the 'monomeric ring' model, cohesin complexes are predicted to exist as monomeric complexes with a core subunit composition ratio of 1:1:1:1. The 'handcuff' model in contrast, proposes that cohesin complexes that mediate cohesion exist as dimeric cohesin rings bridged by a single STAG subunit; that is these complexes exist in a 1:1:1:0.5 subunit ratio, with 0.5 being the stoichiometry of the sum of STAG1 and STAG2 proteins relative to the other three subunits (*Zhang et al., 2008*; *Zhang and Pati, 2009*). Previous experiments have indicated that approximately half of all chromatin-bound cohesin complexes become cohesive during DNA replication (*Gerlich et al., 2006*; *Kueng et al., 2006*; *Schmitz et al., 2007*). Thus, to be consistent with the handcuff model, our measurements should have revealed a 1:1:1:1 stoichiometry for chromatin-bound cohesin before DNA replication and a 1:1:1:0.75 stoichiometry after replication.

A cohesin subunit stoichiometry of 1:1:1:1 has been previously reported for cohesin complexes immunoprecipitated from unfractionated HeLa cells (*Holzmann et al., 2011*). To compare the stoichiometry of soluble and chromatin-bound cohesin, we used SCC1 antibody beads to

immunoprecipitate cohesin from soluble and chromatin fractions isolated from G1, G2 and prometaphase HeLa cells and then subjected the immunoprecipitates to LC-MS analysis on an LTQ Orbitrap instrument. The ratios of SMC1, SMC3 and STAG1/2 relative to SCC1 were below one in all experimental conditions, possibly reflecting a loss of co-precipitating material during sample processing. It is therefore likely that this method underestimates the ratio of cohesin subunits to SCC1. We identified SMC1, SMC3 and STAG1/2 at ratios of 0.89, 0.95 and 0.83 relative to SCC1 in immunoprecipitates from G1 chromatin, at 0.9, 0.97 and 0.85 in G2 and at 0.88, 0.98 and 0.88 in prometaphase (*Figure 1—figure supplements 3* and *4*, *Appendix 1—table 2*). Thus, the stoichiometry of cohesin remains close to 1:1:1:1 in G1, G2 and prometaphase. Importantly, the 95% confidence interval for the ratio of STAG1/2 to SCC1 on G2 chromatin was 0.77–0.93. Since this is likely to be an underestimation of the true STAG1/2:SCC1 ratio, our experiments are consistent with the monomeric ring model for cohesion establishment. However, this approach is unable to distinguish between 1:1:1:1 and 2:2:2:2 subunit ratios. Therefore, we cannot exclude that a fraction of cohesin complexes exists as dimers or multimers, as proposed in the accompanying study (*Cattoglio et al., 2019*).

## Fluorescence correlation spectroscopy analysis of cohesin, CTCF and other cohesin regulators

By measuring the changes in photon counts caused by single molecule fluctuations within a small illumination volume, fluorescence correlation spectroscopy (FCS) allows determination of a number of biophysical parameters, including the concentration of fluorescently-tagged proteins within living cells. To this end, we used HeLa cell lines in which the cohesin subunits SCC1, STAG1, STAG2 and NIPBL, WAPL, sororin, and CTCF were homozygously tagged with enhanced green fluorescent protein (EGFP) at their endogenous loci using CRISPR-Cas9 genome editing (*Figure 2—figure supplement 1*). We synchronised these cell lines in G1 phase, G2 phase or prometaphase using thymidine and nocodazole arrest/release protocols similar to those used in our MS analysis. FCS measurements were automatically acquired from multiple positions in the nucleus and cytoplasm (*Figure 2A*) and protein concentrations were computed (*Figure 2B*, *Appendix 1—table 3*). The G1 concentrations of SCC1 and STAG2 in the nucleus were measured to have a median of 330 nM and 280 nM, respectively. Consistent with our LC-MS data, the concentration of nuclear STAG1 was lower, at around 70 nM. The cytoplasmic concentration of all proteins measured was low and frequently fell below the detection limit of FCS (*Appendix 1—table 3*). To estimate the stoichiometry of our proteins of interest, we compared the fluorescence intensity of the molecules detected by FCS in our EGFP-tagged cell lines to those detected in a cell line that expressed freely diffusing monomeric EGFP (mEGFP). The counts per molecule (CPM) in all our EGFP-tagged cell lines was similar to that of mEGFP,

**Table 2.** FCS quantification of copy number of cohesin subunits and regulators.
Absolute copy number of cohesin subunits and regulators obtained from FCS protein concentration measurements in the nucleus/chromatin and cytoplasmic compartment of cells (*Figure 2—source data 1*). Copy numbers were calculated by multiplying the protein concentrations by the cell cycle-specific volume of the respective cellular compartment and Avagadro's constant as described in Materials and Methods. Missing or italicised numbers indicate that the number of successful FCS measurements was not sufficient to estimate the protein concentration. Note that the EGFP-sororin cell line displayed a mitotic defect, raising the possibility that EGFP-sororin may be hypomorphic. Data are tabulated as the median. The 68% interval of the distribution is listed in brackets.

| | G1 | | G2 | | Prometaphase | |
|---|---|---|---|---|---|---|
| Protein | nucleus/chromatin | cytoplasm | nucleus/chromatin | cytoplasm | nucleus/chromatin | cytoplasm |
| SCC1 | 250755 [160752; 387511] | 10690 [2426; 28277] | 291939 [228902; 547040] | 6279 [2917; 9460] | 52195 [35147; 72881] | 203259 [138338; 303046] |
| STAG1 | 50062 [26338; 95212] | 1738 [419; 6726] | 90712 [59923; 152640] | 1048 [359; 3369] | 10714 [6958; 16682] | 34860 [26096; 50824] |
| STAG2 | 221261 [158694; 291715] | 18168 [5571; 59386] | 281503 [202964; 384802] | 29080 [9410; 92086] | 67163 [46974; 105183] | 249048 [183810; 360561] |
| NIPBL | 146764 [107180; 202180] | 7314 [3462; 13262] | 162915 [101109; 218244] | 10537 [4819; 18812] | 32357 [22248; 44222] | 135313 [114223; 172737] |
| WAPL | 91114 [68317; 115878] | - | 100084 [70256; 125633] | - | 20196 [13253; 30023] | 88677 [68173; 141144] |
| SORORIN | 44396 [21054; 84541] | 1216 [363; 3551] | 104939 [40639; 163976] | 4580 [832; 27180] | 25099 [16846; 38736] | 103462 [61909; 139749] |
| CTCF | 181157 [131295; 259610] | 3671 [1192; 6294] | 206494 [157309; 284756] | *9708 [9256; 10160]* | 51898 [32725; 80216] | 143505 [87355; 243766] |

DOI: https://doi.org/10.7554/eLife.46269.012

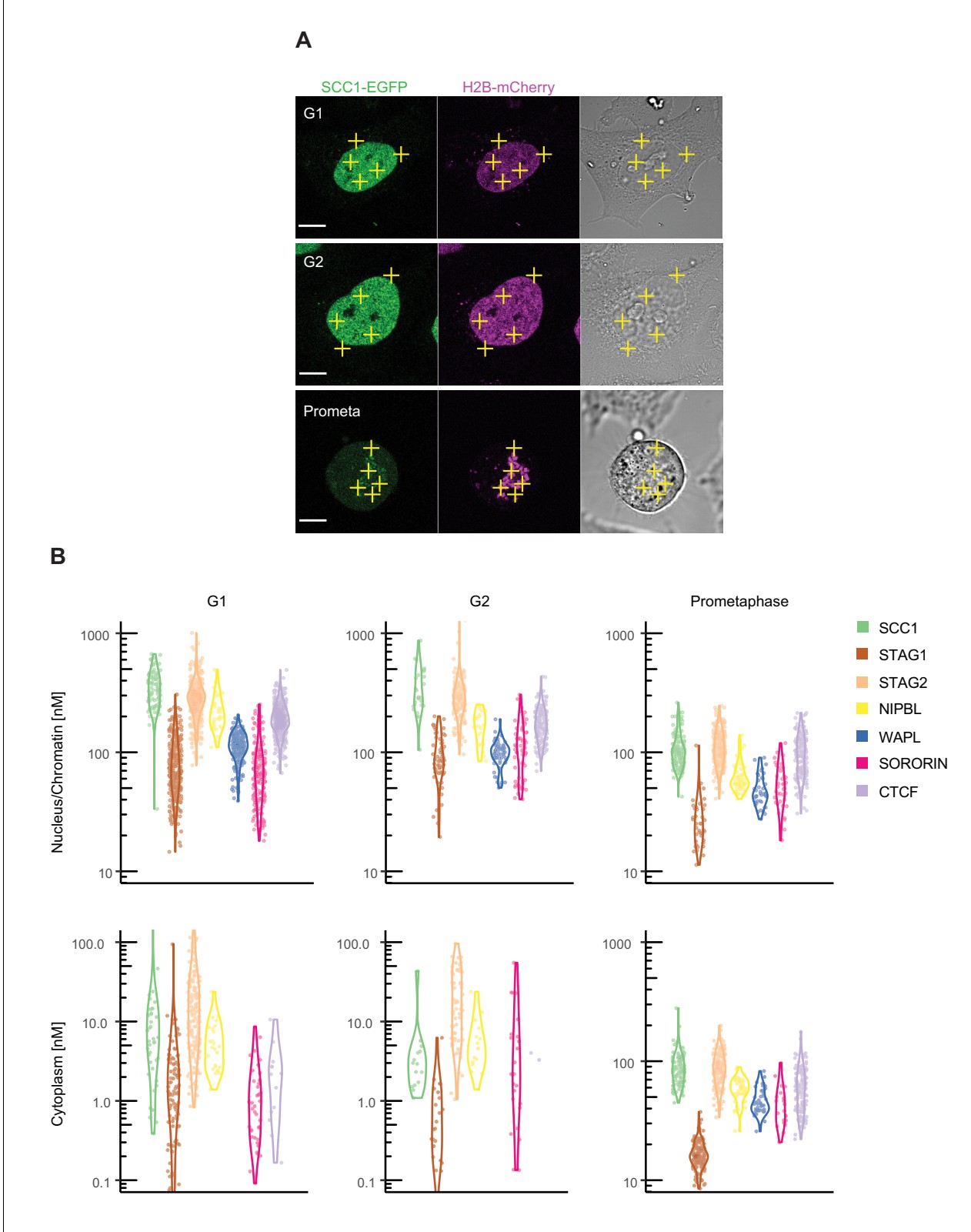

**Figure 2.** Fluorescence correlation spectroscopy of cohesin subunits and regulators. FCS measurements to estimate the concentration of endogenously GFP-tagged proteins in the nucleus/chromatin and cytoplasm of G1, G2 and prometaphase cells. (**A**) FCS measurements were taken at different positions (three in the nucleus/chromatin, two in the cytoplasm) in cells in G1 (top), G2 (middle) or prometaphase (bottom), whereby photon counts were recorded for 30 s at each position. Example images for SCC1-EGFP H2B-mCherry in the GFP (left), DNA (middle) and transmission channels are

*Figure 2 continued on next page*

*Figure 2 continued*

shown. Scale bar: 10 µm. (**B**) Probability density violin plots of the GFP-based protein concentrations determined from each FCS measurement (dots) after data fitting and quality control. Note that the EGFP-sororin cell line displayed a mitotic defect, raising the possibility that EGFP-sororin may be hypomorphic. Data were derived from 2 to 4 experiments per condition, 16–254 cells, and 80–1270 FCS measurements. Data associated with this figure are included in *Figure 2—source data 1*.

DOI: https://doi.org/10.7554/eLife.46269.009

The following source data and figure supplement are available for figure 2:

**Source data 1.** The zip file contains the data used to generate *Figure 2* and *Table 2*, *Appendix 1—tables 5* and *6*.
DOI: https://doi.org/10.7554/eLife.46269.011
**Figure supplement 1.** Cell line characterisation.
DOI: https://doi.org/10.7554/eLife.46269.010

indicating that most cohesin complexes and cohesin regulators exist as monomers (the mean of the median protein of interest: free mEGFP CPM ratios was 1 ± 0.3, *Appendix 1—table 4*). This also indicates that the concentrations obtained using FCS represent the total fraction of diffusing proteins.

We next converted the protein concentration values to absolute copy numbers using cell volumes calculated from 3D segmentation of the nucleus and an estimate of the cell to nucleus volume ratio (*Table 2*). Using this approach, we estimated that approximately 260,000 copies of SCC1-mEGFP, 50,000 copies of STAG1-EGFP and 240,000 copies of STAG2-EGFP exist per G1 cell. This compares to 350,000 copies of SCC1, 130,000 copies of STAG1 and 160,000 copies of STAG2 per G1 cell as estimated by LC-MS. Using bootstrapping, we estimated that the two orthogonal methods agree to within an average factor of 1.5, and that a ratio of one, that is equal protein numbers for the two methods, was within 68% of the error for 6 out of 9 conditions (*Appendix 1—table 5*, see Discussion). As also observed in the LC-MS dataset, the total copy number of cohesin subunits increased in G2 cells compared to G1 cells (*Tables 1* and *2*). The intracellular concentration of cohesin was maintained in G2 despite this increase in total copy number, since the volume of G2 cells also increased (*Figure 2*, *Table 2*, *Appendix 1—table 3*).

We found that NIPBL, WAPL, sororin and CTCF were all sub-stoichiometric relative to SCC1 in G1 and G2 HeLa cells, although the 68% distribution intervals for nuclear SCC1 and NIPBL overlapped in G1 (*Figure 2B* and *Appendix 1—table 3*). Sub-stoichiometry between NIPBL and cohesin is consistent with estimates comparing nuclear levels of fluorescently tagged SCC1 and NIPBL in the human cell line HCT116 (*Rhodes et al., 2017*).

## Dynamics of chromatin-bound cohesin

Our FCS experiments allowed us to measure the number of cohesin complexes that reside in the nucleus but not the number of chromatin-bound complexes, that is those that might actually contribute to chromatin architecture and sister chromatid cohesion. To investigate the dynamics of nuclear cohesin in G1 and G2 phase (*Figure 3—figure supplement 1A*), we performed inverse fluorescence recovery after photobleaching (iFRAP) using the same SCC1-EGFP cell line that was used in our FCS experiments (*Figure 3A,B*). Consistent with previous studies that relied on ectopically expressed

**Table 3.** FCS/iFRAP estimates of soluble, dynamic and stable nuclear SCC1-mEGFP copy number. Copy numbers were calculated by multiplying the median nuclear FCS copy number measurements (*Table 2*) by the average and s.d. population fractions as determined by iFRAP (*Figure 3C*).

| | Copy number | |
| --- | --- | --- |
| | **G1** | **G2** |
| soluble | 91318 ± 10277 | 79239 ± 21371 |
| dynamic | 159437 ± 10277 | 104952 ± 26360 |
| stable | 0 | 107748 ± 26360 |
| total | 250755 | 291939 |

DOI: https://doi.org/10.7554/eLife.46269.015

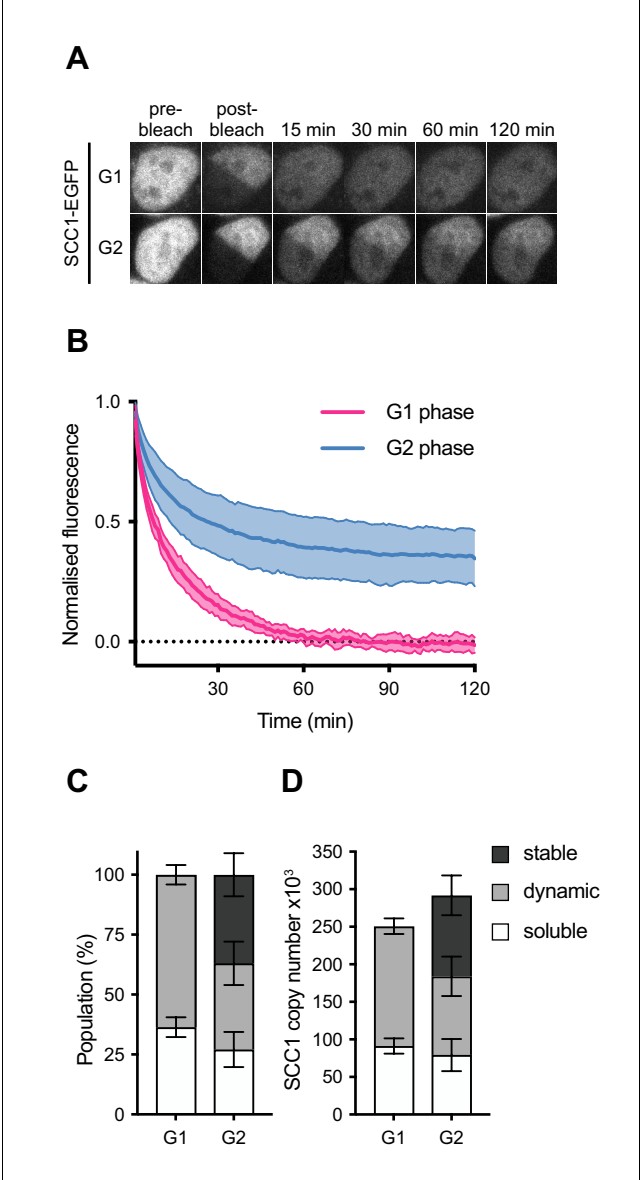

**Figure 3.** Dynamics and distribution of nuclear SCC1-mEGFP in G1 and G2 phase. (**A**) Images of inverse fluorescence recovery after photobleaching (iFRAP) in SCC1-mEGFP cells in G1 and G2 phase (*Figure 3—figure supplement 1A*). Half of the nuclear SCC1-mEGFP fluorescent signal was photobleached and the mean fluorescence in the bleached and unbleached regions was monitored by time-lapse microscopy. (**B**) The difference in fluorescence signals between the bleached and unbleached regions was normalised to the first post bleach image and plotted (mean ± S.D., n = 19 per condition). (**C**) SCC1-mEGFP distribution in the nucleus in G1 and G2 phase. Dynamic and stable populations were estimated using curve fittings from *Figure 3—figure supplement 1B,C*. Soluble populations were estimated by measuring the reduction in signal intensity in the unbleached area after bleaching (*Figure 3—figure supplement 1G*). (**D**) FCS/FRAP estimates of soluble, dynamic and stable nuclear SCC1-mEGFP copy number (see *Table 3* for exact values).

DOI: https://doi.org/10.7554/eLife.46269.013

The following figure supplement is available for figure 3:

**Figure supplement 1.** Dynamics and distribution of nuclear SCC1-mEGFP in G1 and G2 phase.

DOI: https://doi.org/10.7554/eLife.46269.014

GFP-tagged cohesin subunits (*Gerlich et al., 2006*; *Schmitz et al., 2007*; *Ladurner et al., 2014*; *Ladurner et al., 2016*), the recovery kinetics obtained from G1 phase cells could largely be fitted using a single exponential function, indicating cohesin was dynamically bound to chromatin with a residence time of 13.7 ± 2.2 min (*Figure 3—figure supplement 1B,D*). We refer to this population of cohesin as 'dynamic'. Previous work has revealed that it arises from repeated loading and WAPL-mediated release events (*Gandhi et al., 2006*; *Kueng et al., 2006*; *Chan et al., 2012*; *Buheitel and Stemmann, 2013*; *Eichinger et al., 2013*; *Tedeschi et al., 2013*; *Huis in 't Veld et al., 2014*). Also consistent with previous studies, the recovery kinetics obtained from cells in G2 phase could only be fitted using a double exponential function with a dynamic residence time of 10.0 ± 3.1 min and a stable residence time of 8.6 ± 4.1 hr (*Figure 3—figure supplement 1C,D,E* and *Gerlich et al., 2006*).

Since highly mobile proteins diffuse from the unbleached area to the bleached area during the bleaching time period, the reduction in GFP signal in the unbleached area provides an estimate of the soluble fraction of SCC1-EGFP (*Figure 3—figure supplement 1G*). This reduction in GFP signal was not due to general photobleaching, since this signal did not decrease in the nuclei of cells that were not subjected to iFRAP (*Figure 3—figure supplement 1G*, 'unbleached'). According to our FCS analysis, approximately 250,000 copies of SCC1-GFP reside in the nucleus in G1. Our iFRAP analysis allowed us to estimate that approximately 64 ± 4% (160,000 ± 10,000) of these molecules are bound to chromatin (*Figure 3C,D*, *Table 3*). In G2, we estimate that 37 ± 9% of nuclear SCC1-EGFP is bound stably to chromatin, 36 ± 9% is bound dynamically and 27 ± 7% is soluble (*Figure 3C*). Taking into account our FCS measurements of SCC1-EGFP copy number in G2 phase, we estimate that around 108,000 ± 26,000 cohesin molecules are stably bound, 105,000 ± 26,000 bind dynamically and 79,000 ± 21,000 are soluble (*Figure 3D*, *Table 3*). Thus, our LC-MS and FCS estimates of the number of SCC1 molecules bound to chromatin in a HeLa cell agree to within a factor of 2.3 in G1 and a factor of 1.4 in G2 (*Appendix 1—table 1*, *Table 3*). We suspect that this difference is caused by the removal of a fraction of dynamically chromatin-bound cohesin during sample preparation for LC-MS (see Discussion).

## A mathematical model for cohesin binding to chromatin

We observed that the number of chromatin-bound cohesin complexes increased by around a factor of two in G2 cells compared to G1 cells, as measured directly by LC-MS (*Table 1*) and indirectly by integrating our FCS and FRAP data (see above). This increase coincides with the appearance of a stably-bound population of cohesin. To explore whether the observed increase in cohesin's residence time is sufficient to explain the two-fold increase in chromatin-bound cohesin complexes, we performed mathematical modelling. For this we considered the inter-conversion between the different unbound and chromatin-bound forms of nuclear cohesin (*Figure 4A*). By performing a number of algebraic substitutions (see Materials and methods), we generated an equation that allows us to plot the equilibrium distribution of the unbound and dynamically chromatin-bound forms of cohesin as a function of the stably chromatin-bound fraction (*Figure 4B*). The fraction of stable cohesin ($s$) in G2 phase cells is ~0.37 (*Figure 3C*). According to our model, if $s = 0.37$ the chromatin-bound fraction ($b_T$) should be 0.77 (*Figure 4B*). This is very close to the experimentally-determined fraction of chromatin-bound cohesin in G2 (0.73, *Figure 3C*). Therefore, we propose that the only major distinction between cohesin dynamics in G1 and G2 phase cells is that a fraction of cohesin becomes stably bound in G2 phase.

## Implications of absolute cohesin copy numbers for the occupancy of cohesin enrichment sites

Our current knowledge regarding the genomic distribution of human cohesin and its regulators derives largely from population-based ChIP-seq experiments. The distribution of human cohesin on DNA has only been analysed for the 'mappable' non-repetitive part of the genome, and most ChIP experiments that have been performed for this purpose have only revealed the relative distribution of cohesin and can therefore not be used for a quantitative analysis. Nevertheless, it is interesting to compare the absolute number of cohesin complexes that we have measured here with data on cohesin enrichment sites in the human genome. We have identified around 37,000, 35,000 and 47,000 sites for SMC3, STAG1 and STAG2, respectively in the mappable fraction of the human genome in G1-synchronised HeLa cells (*Figure 5A*, *Appendix 1—table 6*). 88% of SMC3 sites overlap with the

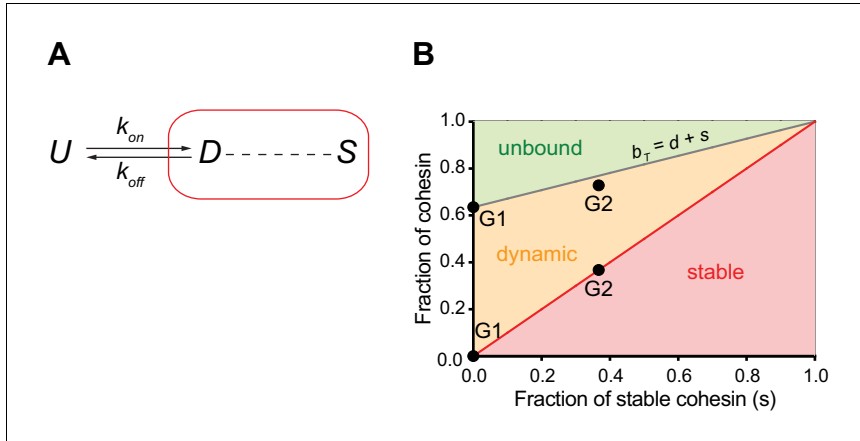

**Figure 4.** Mathematical model of cohesin chromatin binding in G1 and G2 phase. (**A**) Model describing transitions between unbound (U), dynamic (D) and stable (S) forms of nuclear cohesin. The '$k$'s refer to the first order rate constants (with a dimension of time$^{-1}$) of individual transitions. (**B**) Equilibrium distribution of the unbound and dynamic forms of cohesin as a function of the stable fraction. The grey line represents the fraction of nuclear cohesin that is chromatin-bound ($B_T$). The red line represents the fraction of nuclear cohesin that is stably chromatin-bound (S). The black dots represent the experimentally observed values of $B_T$ and S in G1 and G2.
DOI: https://doi.org/10.7554/eLife.46269.016

combined enrichment sites of STAG1 and STAG2, and 77% overlap with CTCF (*Figure 5B*, *Appendix 1—table 7*.).

If we assume that cohesin occupies the HeLa genome (7.9 Mb; *Landry et al., 2013*) with equal frequency as in the 'mappable' human genome (2.7 Mb) there should be around 117,000 cohesin enrichment sites per HeLa cell (~40,000 ÷ 2.7 × 7.9). If one assumes that all ~160,000 dynamically chromatin-bound cohesin complexes identified in a single cell by FCS/FRAP are positioned at the 117,000 cohesin enrichment sites, it is therefore theoretically possible that every site is occupied simultaneously by cohesin in a single G1 cell. However, for reasons explained in the Discussion, at any given time many cohesin complexes may not be positioned at cohesin enrichment sites, in which case not all of these could be simultaneously occupied by cohesin.

We identified around 41,000 CTCF binding sites in ChIP-seq experiments performed in G1-arrested HeLa cells (*Figure 5A*, *Appendix 1—table 7*). Using the same logic as described above, this equates to around 120,000 potential CTCF binding sites per HeLa cell. Using FCS, we estimated that approximately 180,000 copies of GFP-tagged CTCF reside in the nucleus in G1 (*Table 2*). It is unknown what fraction of these molecules is bound to chromatin in HeLa cells, and whether CTCF binds to chromatin as multimers, as has been proposed (*Pant et al., 2004*; *Yusufzai et al., 2004*; *Bonchuk et al., 2015*). Given that we estimate that the number of CTCF molecules in the nucleus is similar to the number of CTCF binding sites, and assuming that at steady state not all CTCF molecules are chromatin bound, it is possible that not every site is occupied simultaneously by CTCF in a single G1 cell (see Discussion).

If cohesin/CTCF enrichment sites represent positions at which cohesin loop extrusion frequently stalls, an estimate as to the average distance a cohesin complex might travel during loop extrusion could be made by determining the genomic distances between pairs of cohesin enrichment sites. If every ChIP-seq peak were distributed equally within the mappable genome, we would expect to detect around one peak every 67.5 kb (2.7 Mb ÷ 40,000). We found that SMC3 enrichment sites were actually distributed over a very broad range of distances, ranging from less than 10 kb apart to greater than 2 Mb (*Figure 5C*). However, around 80% percent of the detected SMC3 enrichment sites resided within 100 kb of each other. We note that this predicted cohesin loop size is very similar to those calculated for the loops formed by the related condensin complex on mitotic chromosomes (*Gibcus et al., 2018*; *Walther et al., 2018*) and is also consistent with the range of loop sizes predicted to exist in interphase chromatin (*Wutz et al., 2017*). As described above, it is possible that not every cohesin enrichment site is occupied simultaneously in a single cell, and the size of a

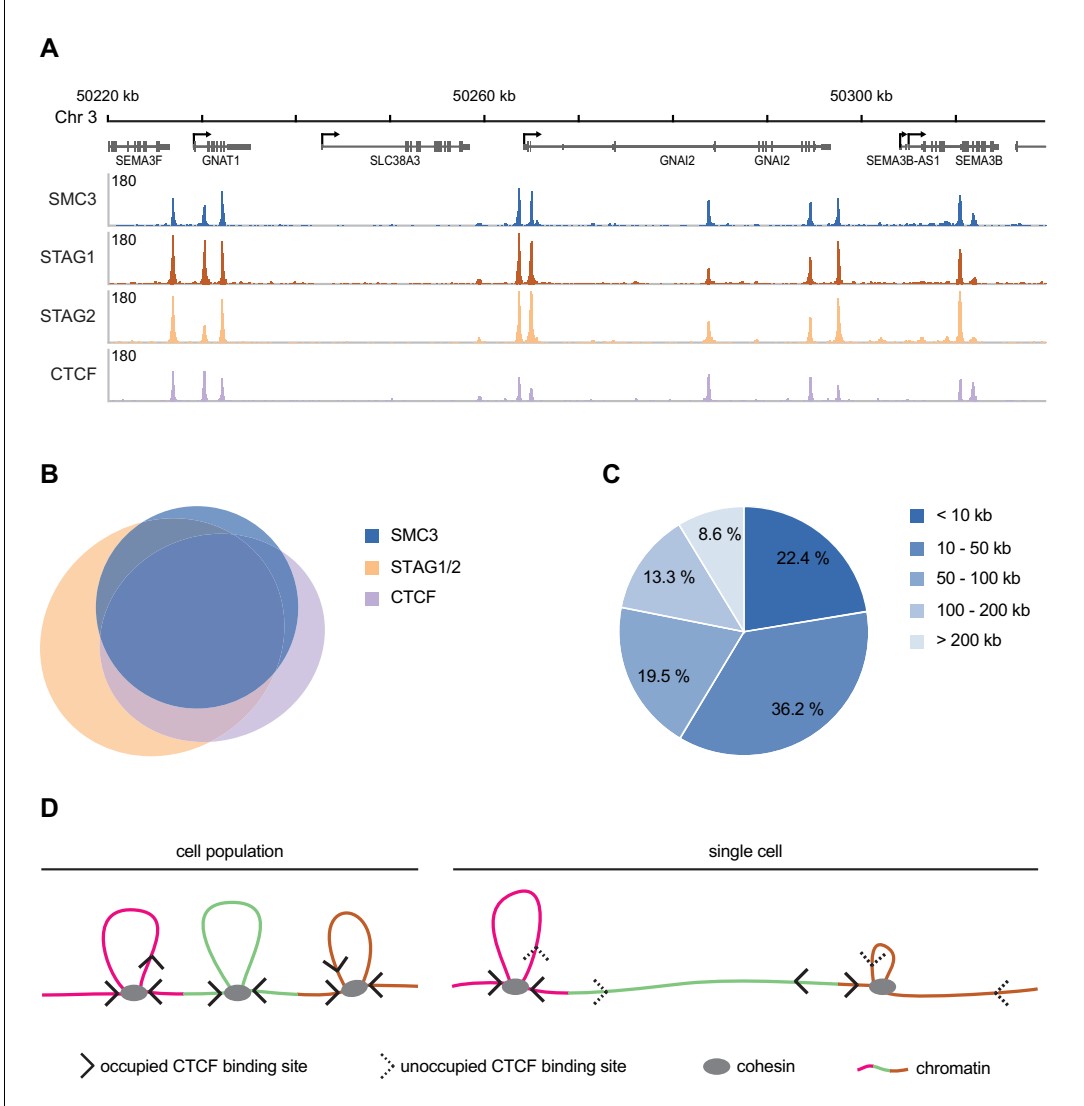

**Figure 5.** Genomic distribution of SMC3, STAG1, STAG2 and CTCF in G1 phase. (**A**) Enrichment profiles of SMC3, STAG1, STAG2 and CTCF along an exemplary 100 kb region of chromosome 3, illustrating typical distribution and co-localisation of sequencing read pileups. Genes within this region are depicted above. SMC3 and CTCF were immunoprecipitated from HeLa Kyoto using anti-SMC3 and anti-CTCF antibodies, respectively. EGFP-STAG1 and STAG2-EGFP were immunoprecipitated from the respective genome-edited cell lines using anti-GFP antibodies. (**B**) Area-proportional threefold eulerAPE Venn diagram (www.eulerdiagrams.org/eulerAPE/) illustrating genome-wide co-localisation between SMC3, CTCF, and the combined set of STAG1 and STAG2 coordinates. (**C**) Pie chart depicting categories of pairwise genomic distances between SMC3 enrichment sites. (**D**) Schematic comparing the occupancy of cohesin and CTCF across a cell population and within a single cell. Incomplete occupation of cohesin and CTCF binding sites can explain why chromatin loops are not uniform and how cohesin can 'skip' past CTCF binding sites.

DOI: https://doi.org/10.7554/eLife.46269.017

loop formed by a cohesin complex might depend on where in between the two adjacent cohesin enrichment sites cohesin initially binds. Nevertheless, this analysis suggests that at least some cohesin complexes might extrude loops around 100 kb in size. This is in line with predictions of loop extrusion processivity based on polymer modelling (120–240 kb; *Fudenberg et al., 2016*).

## Discussion

### A comparison between LC-MS and FCS measurements of cohesin copy number

As mentioned above, our LC-MS and FCS-derived estimates of total cohesin subunit copy number were within an overall factor of 1.3 of one another in G1 and 1.4 in G2 (*Appendix 1—table 5*). Given the major differences between these orthogonal techniques, the similarity between both datasets is remarkable. The estimated copy numbers for STAG1 showed larger differences, however. Several factors, including experimental variability (intra-assay variability for each of the two methods was ~20–30%), could have contributed to this discrepancy (see Materials and methods). Our FCS-derived estimates of chromatin-bound SCC1 were 2.3-fold higher than our LC-MS estimates in G1 and 1.4-fold higher in G2. It is possible that a fraction of dynamically chromatin-bound cohesin complexes was lost during the purification steps that preceded our LC-MS analysis.

### Occupancy of cohesin and CTCF enrichment sites in a single cell

In the loop extrusion model of cohesin-mediated genome organisation, cohesin is predicted to bind to chromatin and translocate to extrude loops (*Nasmyth, 2001*; *Sanborn et al., 2015*; *Fudenberg et al., 2016*). This model therefore predicts that chromatin-bound cohesin must transiently exist at other sites in addition to those identifiable by standard ChIP-seq. Consistently, calibrated ChIP-seq experiments suggest that significant amounts of budding yeast cohesin localises to regions between peaks (*Hu et al., 2015*). Direct evidence of cohesin-mediated loop extrusion is so far missing, however the budding yeast orthologue of the related SMC complex condensin has been shown to translocate unidirectionally along DNA (*Terakawa et al., 2017*) and perform loop extrusion of DNA in vitro (*Ganji et al., 2018*). Fission yeast and vertebrate cohesin can translocate along DNA in vitro (*Davidson et al., 2016*; *Kanke et al., 2016*; *Stigler et al., 2016*) and ChIP-seq studies in bacteria, yeast and mammalian cells have all revealed that cohesin/SMC enrichment sites can vary in their positions (*Lengronne et al., 2004*; *Hu et al., 2015*; *Ocampo-Hafalla et al., 2016*; *Busslinger et al., 2017*; *Wang et al., 2017*; *Petela et al., 2018*; *Wang et al., 2018*). If cohesin is indeed mobile on chromatin, our finding that the number of chromatin-bound cohesin complexes is similar to the number of potential cohesin binding sites means it is unlikely that every cohesin enrichment site is occupied in a single G1 cell. However, if cohesin performs loop extrusion as a monomer, and the cohesin ChIP-seq binding sites that overlap with CTCF represent stalled extrusion events, it could be that a single cohesin complex could occupy the two cohesin enrichment sites present at the base of chromatin loops. This would decrease the number of cohesin complexes required to simultaneously occupy all cohesin enrichment sites in a single cell.

We estimate that there are around 120,000 potential CTCF binding sites and 180,000 copies of CTCF per HeLa cell nucleus. This suggests that if the chromatin-bound fraction of CTCF is below around 0.67, not all CTCF binding sites would be occupied simultaneously. This would provide a potential explanation as to how cohesin might 'skip' past the CTCF sites identified within TADs (*Figure 5D*) and also how longer chromatin loops can form following WAPL depletion despite no detectable change in CTCF distribution (*Gassler et al., 2017*; *Haarhuis et al., 2017*; *Wutz et al., 2017*). Similar conclusions regarding the copy number and occupancy of cohesin and CTCF have recently been drawn in mouse embryonic stem cells and human U2OS cells (*Hansen et al., 2017*; *Cattoglio et al., 2019*). Importantly, these authors concluded that only around 49% of nuclear CTCF molecules are actually chromatin-bound (*Hansen et al., 2017*).

### Cohesin residence time and genome organisation

Our FRAP data indicate that in G1, cohesin is bound to chromatin dynamically with a residence time of around 13 min, although our measurements cannot exclude that there are also more short-lived interactions between cohesin and chromatin as reported earlier (*Ladurner et al., 2014*). To try to understand whether dynamic cohesin complexes might participate in loop extrusion, it is interesting to consider how fast cohesin would have to extrude to generate a chromatin loop of a defined size within its estimated residence time. To generate a 100 kb loop (the genomic distance that covers more than 80% of cohesin enrichment sites; *Figure 5C*), a dynamically chromatin-bound cohesin complex would need to extrude at around 8 kb/min. Since we found that the distance between

neighbouring cohesin enrichment sites was highly variable and cohesin occupancy at these sites is likely to be below 100%, an alternative estimate as to the dimensions of a typical chromatin loop is to use the median TAD size determined in Hi-C experiments (~185 kb, *Rao et al., 2014*). Using this value, dynamically chromatin-bound cohesin would need to extrude at rates of around 14 kb/min. These calculated rates of dynamic cohesin-mediated loop extrusion are close to the predicted rate of loop extrusion by cohesin in HeLa cells (22.5 kb/min, *Rao et al., 2017*), condensin II in chicken cells (6–12 kb/min, *Gibcus et al., 2018*), SMC complexes in *B. subtilis* (54 kb/min, *Wang et al., 2017*) and the maximal rate of loop extrusion observed for yeast condensin in vitro (90 kb/min, *Ganji et al., 2018*).

## Genome organisation and sister chromatid cohesion

The ring model of sister chromatid cohesion proposes that cohesive cohesin complexes topologically entrap replicated sister chromatids *Haering et al. (2008)*. If cohesin extrudes loops of chromatin, it is possible that it does so without topological entrapment, raising the possibility that two populations of cohesin exists in cells, one that is competent for loop extrusion and the other for cohesion (*Srinivasan et al., 2018*). Cohesion is believed to be mediated by stably-bound complexes, which comprises around half of all chromatin-bound cohesin in G2 (*Figure 3C* and *Gerlich et al., 2006*). Depletion of WAPL results in the stabilisation of cohesin complexes on chromatin and a dramatic change in chromatin architecture, indicating that stably-bound cohesin complexes are able to function in chromosome organisation and that alteration of cohesin's residence time may influence this process (*Kueng et al., 2006*; *Tedeschi et al., 2013*; *Gassler et al., 2017*; *Haarhuis et al., 2017*; *Wutz et al., 2017*). However, the number and position of TADs and loops does not differ significantly between G1 and G2 cells (*Wutz et al., 2017*), that is chromatin architecture does not detectably change even though many cohesin complexes are stably bound to chromatin in G2. This raises the interesting possibility that stably-bound cohesive cohesin in G2 is functionally distinct from the stably-bound cohesin found in cells depleted of WAPL. If so, it is possible that this is because the former might interact with two stretches of chromatin topologically and the latter might do so non-topologically.

Considering the notion that around half of chromatin-bound cohesin is stably-bound in G2 and may not function in loop extrusion, we were interested to use our LC-MS, FCS and FRAP data to compare the number of dynamically chromatin-bound cohesin complexes in cells synchronised in G1 and G2. If chromatin loops are formed by dynamically bound cohesin one might expect that more of these complexes are present on chromatin in G2-phase to be able to form long-range chromatin interactions on both sister chromatids. However, even though the total number of cohesin complexes bound to chromatin in G2 is twice of that in G1, our FRAP data indicate that the number of dynamically chromatin-bound cohesin complexes is actually reduced (*Tables 1*, *2* and *3*, and *Figure 3D*). Thus, either stably-bound cohesin participates in genome organisation in some way – without any of the changes in chromatin structure observed following WAPL depletion – or the two-fold increase in DNA content in G2 must be organised by relatively fewer cohesin complexes.

## Implications of cohesin copy number measurements for cohesion at centromeric regions

Using FCS and iFRAP, we estimated that around 213,000 cohesin complexes are bound to chromatin in a typical G2 cell, of which around half are bound stably to chromatin (*Table 3*, *Figure 3D*). Since the stable fraction is the one thought to mediate cohesion (*Gerlich et al., 2006*), this suggests that around 108,000 cohesin complexes establish cohesion between replicated genomes that each contain 117,000 predicted cohesin enrichment sites. Thus, if cohesion is mediated by monomeric cohesin, it is unlikely that it is established at every cohesin enrichment site in a single cell. This is consistent with the finding that proximity to a cohesin enrichment site does not correlate with proximity between sister chromatids in G2 (*Stanyte et al., 2018*).

Our results obtained with prometaphase arrested HeLa cells imply that the situation may be different at centromeres. Using LC-MS, we estimated that only around 14,000 cohesin complexes were bound to chromosomes in these cells (*Figure 1* and *Table 1*). These numbers are in good agreement with the previous observation that approximately 90% of all chromatin-bound cohesin complexes dissociate from chromosomes in prophase and prometaphase (*Waizenegger et al., 2000*;

*Gerlich et al., 2006*). Since previous immunofluorescence microscopy experiments have shown that in prometaphase-arrested cells cohesin is predominantly detected at centromeres (*Giménez-Abián et al., 2004*) and because ChIP experiments could not detect cohesin in the non-repetitive part of the genome in such cells (*Wendt et al., 2008*), we assume that most of the 14,000 cohesin complexes that we found on prometaphase chromosomes must have represented cohesin at centromeres. Since the HeLa cells used in this study contained on average 64 chromosomes (*Landry et al., 2013*), our measurements indicate that each of these chromosomes might contain around 200 cohesin complexes in its centromeric region.

With this study, we have measured the number of cohesin complexes and cohesin regulatory proteins in a human cell line that is widely used for studying chromatin organisation and mechanisms of sister chromatid cohesion. We have approximated how many of those cohesin complexes may participate in these two functions and have related this to the genomic distribution of cohesin enrichment sites throughout the cell cycle. Our findings suggest that at least some but possibly many of the predicted cohesin enrichment sites are unoccupied in a single cell at a given timepoint, implying that our current models of chromatin organisation and sister chromatid cohesion derived from large populations of cells may not fully reflect the situation in individual cells.

# Materials and methods

## Key resources table

| Reagent type (species) or resource | Designation | Source or reference | Identifiers | Additional information |
|---|---|---|---|---|
| Cell line (H. sapiens) | HK CRISPR SCC1-mEGFP + H2B-mCherry | parental genome-edited cell line from *Davidson et al. (2016)* | | |
| Cell line (H. sapiens) | HK CRISPR EGFP-STAG1 | this paper | | See Materials and methods subsection 'Generation of EGFP-tagged NIPBL and STAG1 HeLa Kyoto cell lines' |
| Cell line (H. sapiens) | HK CRISPR STAG1-EGFP H8 + H2B-mCherry | parental genome-edited cell line from *Cai et al. (2018)* | | |
| Cell line (H. sapiens) | HK CRISPR STAG2-EGFP F2 + H2B-mCherry | parental genome-edited cell line from *Cai et al. (2018)* | | |
| Cell line (H. sapiens) | HK CRISPR EGFP-NIPBL F1 + H2B-mCherry | this paper | | See *Figure 2—figure supplement 1* and Materials and methods subsection 'Generation of EGFP-tagged NIPBL and STAG1 HeLa Kyoto cell lines' |
| Cell line (H. sapiens) | HK CRISPR EGFP-WAPL H3 + H2B-mCherry | parental genome-edited cell line from *Ladurner et al. (2016)*; *Cai et al. (2018)* | | See *Figure 2—figure supplement 1* |
| Cell line (H. sapiens) | HK CRISPR EGFP-SORORIN D3 + H2B-mCherry | parental genome-edited cell line from *Ladurner et al. (2016)* | | See *Figure 2—figure supplement 1* |
| Cell line (H. sapiens) | HK CRISPR CTCF-EGFP F2 | *Wutz et al. (2017)*; *Cai et al. (2018)* | | |
| Cell line (H. sapiens) | HK H2B-mCherry | *Neumann et al. (2010)* | | |
| Antibody | Rabbit polyclonal Anti-SMC3 | Peters laboratory | Antibody ID:k727 | ChIP, *Figure 5* |
| Antibody | Rabbit polyclonal Anti-CTCF | Merck | Cat #:07–729 RRID:AB_441965 | ChIP, *Figure 5* |

*Continued on next page*

*Continued*

| Reagent type (species) or resource | Designation | Source or reference | Identifiers | Additional information |
|---|---|---|---|---|
| Antibody | Rabbit polyclonal Anti-WAPL | Peters laboratory | Antibody ID:A1017 | Western blotting (1:1000), *Figure 2—figure supplement 1* |
| Antibody | Rabbit polyclonal Anti-sororin | Peters laboratory | Antibody ID:A953 | Western blotting (1:1000), *Figure 2—figure supplement 1* |
| Antibody | Rat polyclonal Anti-NIPBL | Absea | Cat #:010702F01 | Western blotting (1:1000), *Figure 2—figure supplement 1* |
| Antibody | Mouse monoclonal Anti-tubulin | Sigma | Cat #:T-5168 RRID:AB_477579 | Western blotting (1:50000), *Figure 2—figure supplement 1* |
| Antibody | Rabbit polyclonal Anti-GFP | Abcam | Cat #:ab290 RRID:AB_303395 | ChIP, *Figure 5* |

## Data reporting and accessibility

No statistical methods were used to predetermine sample size. The experiments were not randomised. The investigators were not blinded to allocation during experiments and outcome assessment. The mass spectrometry proteomics data have been deposited to the ProteomeXchange Consortium via the PRIDE (*Perez-Riverol et al., 2019*) partner repository with the dataset identifier PXD012712. The FCS data with the autocorrelation curves used to compute protein concentrations are included in *Figure 2—source data 1*.

## Generation of EGFP-tagged NIPBL and STAG1 HeLa Kyoto cell lines

The EGFP-NIPBL and EGFP-STAG1 cell lines were generated by homology-directed repair using CRISPR Cas9(D10A) paired nickase (*Ran et al., 2013*). A donor plasmid comprising homology arms (700–800 bp (NIPBL) and 1300–1500 bp (STAG1) on either side of the coding sequence start site) and EGFP were cloned into plasmid pJet1.2 (Thermo Scientific, K1232). Cas9 guide RNA sequences were identified using the website crispr.mit.edu (NIPBL guide A: gTCCCCGCAAGAGTAGTAAT; NIPBL guide B: gGTCTCACAGACCGTAAGTT; STAG1 guide A: gACAATACTTACTGTAACAC; STAG1 guide B: gTATTTTTTAAGGAAAATTT) and inserted into plasmid pX335 (a gift from Feng Zhang, Addgene, 42335). HeLa Kyoto cells (*Landry et al., 2013*) were transfected with donor, Cas9 nickase plasmids and Lipofectamine 2000 (Invitrogen, 11668019). Media were replaced the next day and cells were maintained for 7 days before sorting GFP positive cells by flow cytometry into 96 well plates. EGFP-NIPBL clone F1 and EGFP-STAG1 clone H7 were selected after verification of homozygous GFP insertion by PCR of genomic DNA, immunoblotting and inspection by microscopy.

## Cell line characterization

All cell lines were free from detectable mycoplasma contamination and have been authenticated by STR fingerprinting (Vienna Biocenter Core Facilities). PCR of genomic DNA was used to verify homozygous GFP insertion in all EGFP-tagged HeLa Kyoto cell lines. The following primers were used: EGFP-NIPBL: ATCGTGGGAACGTGCTTTGGA, GCTCAGCCTCAATAGGTACCAACA. EGFP-WAPL: TGATTTTTCATTCCTTAGGCCCTTG, TACAAGTTGATACTGGCCCCAA. EGFP-sororin: GCTAGCCC TACGTCACTTCC, TGCAGTCCCAGTACACAACG.

Western blotting was used to detect proteins of interest in HeLa Kyoto w.t. and GFP-tagged cell lines. For western blotting of GFP-tagged NIPBL and WAPL HeLa Kyoto cell lines, cells were resuspended in RIPA buffer (50 mM Tris pH 7.5, 150 mM NaCl, 1 mM EDTA, 1% NP-40, 0.5% Na-deoxycholate and 0.1% SDS), supplemented with pepstatin, leupeptin and chymostatin (10 µg/ml each) and PMSF (1 mM). Protein concentration was determined using the Bradford Protein Assay (Bio-Rad Laboratories). Samples were separated by SDS-PAGE and western blotting was performed using the

antibodies described below. For western blotting of GFP-tagged sororin HeLa Kyoto cell line, a chromatin extract was prepared as described previously (*Ladurner et al., 2014*). Antibodies used: rabbit anti-WAPL (Peters laboratory ID A1017), rabbit anti-sororin (Peters laboratory ID A953), rat anti-NIPBL (Absea 010702F01) and mouse anti-tubulin (Sigma, T-5168).

## iFRAP

For live-cell imaging, cells were seeded into LabTek II chambered coverslips (ThermoFisher Scientific) in cell culture medium without riboflavin and phenol red, and cultured at 37°C and 5% $CO_2$ during imaging. Cells in G1 and G2 phase were identified by nuclear and cytoplasmic distribution of DHB-mKate2 signals, respectively. 1 μg/ml cycloheximide was added to the imaging medium 1 hr before the imaging to reduce new synthesis of SCC1-mEGFP. Both FRAP and iFRAP experiments were performed using an LSM880 confocal microscope (Carl Zeiss) with a 40 × 1.4 NA oil DIC Plan- Apochromat objective (Zeiss). Photobleaching was performed in half of nuclear regions with 2 iterations of 488 nm laser at max intensity after acquisition of two images. Fluorescence was measured in bleached- and unbleached regions followed by background subtraction with 1 min interval. iFRAP curves were normalised to the mean of the pre-bleach fluorescent intensity and to the first image after photobleaching. Curve fitting was performed with single exponential functions $f(t)=EXP(-kOff1*t)$ or double exponential functions $f(t)=a*EXP(-kOff1*t)+(1-a)*EXP(-kOff2*t)$ in R using the minpack.lm package (version 1.2.1). Dynamic and stable residence times were calculated from $1/kOff1$ and $1/kOff2$ respectively. Double exponential curve fitting was performed under constraint that $1/kOff2$ is in range between 1.5 hr and 15 hr. Soluble fractions were estimated by the reduction of fluorescence signals in unbleached area after photobleaching.

## Chromatin immunoprecipitation and Illumina sequencing

Cells were synchronised in G1 phase using the same procedure as described for LC-MS. ChIP was performed as described (*Wendt et al., 2008*). Ten million cells were used for one ChIP experiment. Cells were crosslinked with 1/10 medium volume of X-link solution (11% formaldehyde, 100 mM NaCl, 0.5 mM EGTA, 50 mM Hepes pH 8.0) at room temperature for 10 min and subsequently quenched with 125 mM glycine for 5 min. Cells were washed with PBS and collected by mechanical scraping and pelleted by centrifugation. Cell pellets were lysed in lysis buffer (50 mM Tris-HCl pH 8.0, 10 mM EDTA pH 8.0, 1% SDS, protease inhibitors) on ice for 20 min. The DNA was sonicated for 6 cycles (30 sec on/off) using a Biorupter. Ten volumes of dilution buffer (20 mM Tris-HCl pH 8.0, 2 mM EDTA pH 8.0, 1% Triton X-100, 150 mM NaCl, 1 mM PMSF) was added to the lysate, which was then pre-cleared using 100 μl Affi-Prep Protein A beads at 4°C. Immunoprecipitation was performed with rabbit IgG or specific antibody overnight; Affi-Prep Protein A beads were then added for a further 3 hours. Anti-GFP antibody ab290 (Abcam, United Kingom) was used to immunoprecipitate EGFP-STAG1 and STAG2-EGFP. An antibody raised against peptide CEMAKDFVEDDTTHG, Peters lab antibody ID: k727, was used to immunoprecipitate SMC3. Anti-CTCF antibody 07-729 (Merck, Germany) was used to immunoprecipitate CTCF. Beads were washed twice with Wash buffer 1 (20 mM Tris-HCl pH 8.0, 2 mM EDTA pH 8.0, 1% Triton X-100, 150 mM NaCl, 0.1% SDS, 1 mM PMSF), twice with Wash buffer 2 (20 mM Tris-HCl pH 8.0, 2 mM EDTA pH 8.0, 1% Triton X-100, 500 mM NaCl, 0.1% SDS, 1 mM PMSF), twice with Wash buffer 3 (10 mM Tris-HCl pH 8.0, 2 mM EDTA pH 8.0, 250 mM LiCl, 0.5% NP-40, 0.5% deoxycholate), twice with TE buffer (10 mM Tris-HCl pH 8.0, 1 mM EDTA pH 8.0), and eluted twice with 200 μl elution buffer (25 mM Tris-HCl pH 7.5, 5 mM EDTA pH 8.0, 0.5% SDS) by shaking at 65°C for 20 min. The eluates were treated with RNase-A at 37°C for 1 hour and proteinase K at 65°C overnight. Addition of 1 μl glycogen (20 mg/ml) and 1/10th volume sodium acetate (3 M, pH 5.2) was followed by extraction with phenol/chloroform/isoamyl alcohol (25:24:1) and precipitation with ethanol. DNA was resuspended in 100 μl of $H_2O$, and ChIP efficiency was quantified by quantitative PCR (qPCR). The DNA samples were submitted to Vienna BioCenter Core Facilities for library preparation and Illumina deep sequencing.

## ChIP-seq peak calling and site overlap counting

Illumina sequencing results of ChIPseq experiments were mapped against the human hg19 reference assembly using bowtie2 (bowtie-bio.sourceforge.net/bowtie2/index.shtml); the resulting alignments from two biological replicate experiments for each immunoprecipitation were combined as BAM

files using samtools merge (samtools.sourceforge.net/). Peaks were called by MACS 1.4.2 (liulab.dfci.harvard.edu/MACS/) with a P-value threshold of 1e-10 using sample and control inputs. Peak overlaps were calculated by using multovl 1.3 (github.com/aaszodi/multovl). Since occasionally two peaks from one dataset overlap with a single peak in another dataset, the output of such an overlap is displayed as a connected genomic site and counted as one single entry. Consequently, the overall sum of peak counts is reduced when displayed in overlaps.

## LC-MS methods

### Cell culture

HeLa cells were cultured as previously (*Nishiyama et al., 2010*). Cells were synchronised in G2 phase by a double thymidine block (24 hr block in 2 mM thymidine, 8 hr release and 16 hr block in 2 mM thymidine) followed by a 6 hr release into G2 phase. Cells were synchronised in Prometaphase by a double thymidine block followed by a 6 hr release and a 4 hr block in Prometaphase using a final concentration of 100 ng/ml nocodazole. Cells were synchronised in G1 phase using the same procedure as described for Prometaphase cells, but after mitotic shake off, cells were washed twice and cultured for a further 6 hr. Cells were counted using a CASY counter (Schärfe, Germany) and cell counts were verified by manual counting.

### Preparation of soluble and chromatin extracts

two $\times$ $10^7$ HeLa cells in G1, G2 and Prometaphase were re-suspended in 0.5 ml lysis buffer (20 mM Hepes pH 7.6, 150 mM NaCl, 10% glycerol, 0.2% NP40, 1 mM NaF, 1 mM sodium butyrate, 1 mM EDTA and 10 µg/ml (w:v) each of leupeptin, pepstatin and chymostatin) and cells were lysed with 20 strokes using a dounce homogenizer. Chromatin and soluble fractions were separated by centrifugation at 1000 g for 3 min at 4°C. The soluble supernatant was centrifuged for a further 20 min at 20000 g at 4°C (soluble extract). The chromatin pellet was washed by resuspension in 1 ml lysis buffer and centrifugation at 1000 g for 3 min at 4°C. Washing was repeated for a total of 10 times. The chromatin pellet was then re-suspended in 250 µl nuclease buffer (lysis buffer complimented with a final concentration of 0.04 units/µl micrococcal nuclease, 0.1 mg/ml RNase A, 20 mM $CaCl_2$ and 0.04 µl Turbo DNase per µl nuclease buffer), incubated for 2 hr at 4°C and for 15 min at 37°C and finally centrifuged at 20000 g for 5 min (chromatin extract). 90% of soluble and chromatin extracts were used for immunoprecipitation and 10% (corresponding to 2 $\times$ $10^6$ cells) were precipitated using acetone. To compensate for losses during acetone precipitation the protein concentration was measured before and after precipitation using Bradford reagent (on average 17% loss). The protein pellets were resuspended in 1 ml 500 mM tetraethylammonium chloride (TEAB, Fluka) (soluble extract) and 50 µl 500 mM TEAB (chromatin extract), respectively. Proteolysis of soluble and chromatin total cell extracts was performed using a double digest protocol. After reduction in 1 mM tris(2-carboxyethyl)phosphine (TCEP) at 56°C for 30 min and alkylation in 2 mM methyl methanethiosulfonate (MMTS, Fluka) for 30 min, proteins were digested with 500 ng LysC per 20 µl extract (Wako, Richmond, VA) at 37°C for 4 hr. Proteins were then digested with 500 ng trypsin per 20 µl extract (MS grade trypsin gold, Promega) for 16 hr at 37°C.

### Absolute quantification of cohesin in total cell extracts using SRM on 5500 QTRAP

Immediately before LC-SRM analysis, digested soluble and chromatin cell extracts were labelled with the light version of the mTRAQ reagent according to the manufacturer's instructions. For quantification in total soluble extracts, 1.5 µg (experiment 1) and 2 µg (experiment 2) of HeLa extract were used. The mTRAQ light-labelled extract was spiked with heavy labelled reference peptides (2.5 and 5 fmol for the soluble extract and 10 fmol for the chromatin extract). Samples were then separated on a Dionex Ultimate 3000 RSLCnano-HPLC equipped with a C18 PepMap100 column (75 µm ID $\times$500 mm length, 3 µm particle size, 100 Å pore size) (Dionex, Amsterdam, The Netherlands) using the following gradient of solvents A (2% ACN, 0.1% FA) and B (80% ACN, 10% TFE, 0.08% FA) at a flow rate of 250 nl/min: from 2%B to 40% B over 300 min. The mass spectrometer was operated in scheduled SRM mode with the following parameters: multiple reaction monitoring (MRM) detection window of 360 s, target scan time of 2.5 s, curtain gas of 20, ion source gas 1 of 15,

declustering potential of 75, entrance potential of 10. Q1 and Q3 were set to unit resolution. The pause between mass ranges was set to 2.5 ms. Three SRM transitions per peptide were monitored.

## Immunoprecipitation of cohesin complexes

Immunoprecipitation (IP) was performed as described (*Holzmann et al., 2011*). In brief, extracts were incubated on a rotary shaker with 30 µl SCC1 antibody-conjugated beads for 2 hr at 4°C (antibody raised against peptide FHDFDQPLPDLDDIDVAQQFSLNQSRVEEC; Peters lab antibody ID: A890, k575). Beads were then collected by centrifugation and washed three times with 30 beads volume lysis buffer and three times with 30 beads volume lysis buffer minus detergent and protease inhibitor. Finally, beads were washed once with 30 bead volumes of 5 mM Hepes pH 7.8 and 150 mM NaCl. 10 µl of beads were used for elution with 0.2 M glycine pH 2.0 and analysed using SDS-PAGE. 20 µl of beads were re-suspended with 40 µl 500 mM TEAB and subjected to protease elution essentially as described (*Holzmann et al., 2011*). Proteolysis was performed using a double digest protocol using LysC and trypsin (*Holzmann et al., 2011*).

## Analysis of shotgun proteomics data

For peptide identification, the RAW-files were loaded into Proteome Discoverer (version 2.1.0.81, Thermo Scientific). All MS/MS spectra were searched using MS Amanda (Search Engine Version 2.2.6.11097) (*Dorfer et al., 2014*). RAW-files were searched against the human swissprot database (2017-04-02; 20.153 sequences; 11,315.842 residues), using the following search parameters: the peptide mass tolerance was set to 10 ppm and the fragment mass tolerance to 0.8 Da. Trypsin was specified as the proteolytic enzyme, cleaving after lysine and arginine except when followed by proline. The maximal number of missed cleavages was set to 2. Beta-methylthiolation on cysteine was set as fixed and oxidation on methionine was set as variable modification. Proteins were grouped applying a strict parsimony principle and filtered to 1% false discovery rate (FDR) on PSM and protein level using the Percolator algorithm (*Käll et al., 2007*) as integrated in Proteome Discoverer. Proteins identified by a single spectra were removed. In all six samples combined (soluble and chromatin-bound cohesin from cells in G1, G2 and prometaphase), we identified 377 and 265 different proteins in two independent experiments (*Figure 1—source data 1*). In both experiments, core cohesin subunits were among the 14 most abundant proteins identified according to the number of peptide spectrum matches.

## Absolute quantification of purified cohesin using SRM on 5500 QTRAP

Immediately before LC-SRM analysis, digested cohesin was labelled with the light version of the mTRAQ reagent according to the manufacturer's instructions. Labelling efficiency was checked by LC-MS experiments on Orbitrap and found to be >98%. mTRAQ light-labelled cohesin was spiked with 10 fmol (biological experiment 1) and 15 fmol (biological experiment 2), respectively of mTRAQ heavy labelled reference peptides. Preparation of heavy reference peptides was performed essentially as described (*Holzmann et al., 2011*), but peptide EQLSAER was replaced by ELAETEPK. To remove excess of 2-propanol samples were concentrated in a Speed Vac for 10 min to a final volume of approximately 25% of the starting volume and re-diluted with 0.1% trifluoroacetic acid (TFA, Pierce). Samples were then separated on a Dionex Ultimate 3000 RSLCnano-HPLC equipped with a C18 PepMap100 column (75 µm ID ×500 mm length, 3 µm particle size, 100 Å pore size) (Dionex, Amsterdam, The Netherlands) using the following gradient of solvents A (2% ACN, 0.1% FA) and B (80% ACN, 10% TFE, 0.08% FA) at a flow rate of 250 nl/min: from 2%B to 40% B over 120 min. Peptides eluting from the nanoLC were analysed on a 5500 QTRAP instrument (ABSCIEX, Foster City, CA) equipped with a nano-electrospray source with an applied voltage of 2.3 kV. The mass spectrometer was operated in scheduled SRM mode with the following parameters: MRM detection window of 180 s, target scan time of 1.5 s, curtain gas of 20, ion source gas 1 of 15, declustering potential of 75, entrance potential of 10. Q1 and Q3 were set to unit resolution. Pause between mass ranges was set to 2.5 ms. Three SRM transitions per peptide (*Appendix 1—table 1*) were selected and optimised for collision energy by direct infusion of heavy reference peptides. Collision cell exit potentials (CXP) were calculated by dividing Q3 mass by a factor of 29.

## SRM data analysis

SRM data were analysed in Skyline (version 2.5.0.6157). Peptides were quantified based on the height of the elution apex to prevent incomplete quantification due to partially covered elution peaks within the scheduled measurements. Transitions were manually reviewed, and low-quality transitions retracted from subsequent quantification. Soluble SMC1 quantification was not possible since the single SMC1 peptide was filtered out during analysis. Quantitative results were further analysed in R (version 3.4.3).

## **FCS methods**

### Cell culture

HeLa Kyoto (HK) cells (RRID: CVCL_1922) were a gift from S. Narumiya (Kyoto University, Kyoto, Japan [*Landry et al., 2013*]) and grown in 1x high-glucose DMEM (Thermo Fisher Scientific; #41965039) supplemented with 10% (v/v) FBS (Thermo Fisher Scientific; #10270106; qualified, European Union approved, and South American origin), 1 mM sodium pyruvate (Thermo Fisher Scientific; #11360070), 2 mM L-glutamine (Thermo Fisher Scientific; #25030081) and 100 U/mL penicillin-streptomycin (Thermo Fisher Scientific; #15140122) at 37°C and 5% $CO_2$ in 10 cm cell culture dishes (Thermo Fisher Scientific). Cells were passaged every two to three days by trypsinization using 0.05% Trypsin-EDTA (Thermo Fisher Scientific; #25300054) at a confluency of 70–90%.

### Generation of stably expressing H2B-mCherry cells

To generate cells stably expressing H2B-mCherry as DNA marker, cells were transfected with a plasmid encoding H2B-mCherry. In brief, 2 µg of pH2B-mCherry plasmid DNA was incubated with 200 µL of jetPRIME buffer and 4 µL of jetPRIME (Polyplus Transfection; #114–07) for 15 min before addition to cells grown to 80% confluency in one well of a Nunc 6-well plate (Thermo Fisher Scientific; #140685) containing 2 mL of complete cell culture medium. After 4 hr the transfection mix was changed to complete cell culture medium. Cells stably expressing H2B-mCherry were selected with 0.5 µg/mL puromycin (InvivoGen; #ant-pr-1).

### Cell preparation for FCS experiments

For each FCS experiment, HK wild-type (WT) cells were seeded together with HK cells homozygously expressing the EGFP-tagged protein of interest (POI) into individual wells of a Nunc 6-well plate (Thermo Fisher Scientific; #140685) at a concentration of $2 \times 10^5$ cells per well and grown overnight in a cell culture incubator. Alternatively, HK WT cells stably expressing H2B-mCherry (*Neumann et al., 2010*) were seeded together with genome-edited cell lines additionally expressing H2B-mCherry. On the next morning, 2 mM thymidine (Sigma-Aldrich; #T1895) in complete cell culture medium was added per well to arrest cells at the G1/S boundary. After 24 hr the thymidine block was released by washing the cells three times with pre-warmed D-PBS. Directly after release from the G1/S block, cells were trypsinized and $8 \times 10^3$ to $2 \times 10^4$ cells were seeded into individual wells of a 96-well glass bottom plate (zell-kontakt; #5241–20) or a Nunc 8-well LabTek #1.0 chambered coverglass (Thermo Fisher Scientific; #155411). HK WT cells in one well were transiently transfected with a plasmid encoding free mEGFP (pmEGFP-C1; Addgene plasmid #54759, kindly provided by J Lippincott-Schwartz) using FuGENE6 Transfection Reagent (Promega; #E2693) according to the manufacturer's instructions, while HK WT cells in another well remained untransfected. 8 hr after release from the first G1/S block, 2 mM thymidine in complete cell culture medium was added per well. After 16 h cells were released from the second thymidine block as described before by washing with D-PBS and adding complete cell culture medium.

FCS measurements of cells in G2 and G1 phases were performed in the same experiment at time windows 6–9 hr and 14–20 hr, respectively, after release from double thymidine arrest. In preparation for G2/G1 phase experiments, cells were washed with D-PBS and 350 µL imaging medium ($CO_2$-independent imaging medium without phenol red; custom order based on #18045070 from Thermo Fisher Scientific; supplemented with 20% v/v FBS (Thermo Fisher Scientific; #10270106; qualified, European Union approved, and South American origin), 1 mM sodium pyruvate (Thermo Fisher Scientific; #11360070) and 2 mM L-glutamine (Thermo Fisher Scientific; #25030081)) was added per well. For cells not expressing H2B-mCherry as DNA marker, the imaging medium contained 50 nM SiR-DNA (Spirochrome; #SC007; [*Lukinavičius et al., 2015*]).

For FCS experiments of prometaphase cells, 6 hr after release from double thymidine arrest, the complete cell culture medium was changed to imaging medium as described for G2/G1 experiments and additionally supplemented with 330 nM nocodazole (Sigma-Aldrich; #SML1665). FCS measurements of cells in prometaphase arrest were performed 2–15 hr after addition of nocodazole.

## FCS measurements

FCS measurements and fluorescence images were recorded on a Zeiss LSM780, Confocor3, laser scanning microscope equipped with a fluorescence correlation setup and a temperature control chamber. Imaging was performed at 37°C and using a C-Apochromat UV-visible-IR 40X/1.2-NA water objective lens (Zeiss). Data acquisition was performed either manually or with an automatic workflow by using ZEN 2012 Black software (Zeiss) and the software described in *Politi et al. (2018)*. An in-house-designed objective cap and a water pump enabled automatic water immersion during data acquisition.

The effective confocal volume was determined using a 50 nM fluorescent dye solution containing an equimolar mix of Alexa Fluor 488 (Thermo Fisher Scientific; #A20000) and Alexa Fluor 568 (Thermo Fisher Scientific; #A20003). The dye solution was excited with the 488 nm laser (laser at 0.6% excitation (exc.) power) and the 561 nm laser (laser at 0.15% exc. power) and photon counts were recorded for 30 s and six repetitions using two avalanche photodiode detectors (APD). The band pass filters (BPs) were set to 505–540 nm and 600–650 nm, respectively. For FCS measurements of cells homozygously expressing EGFP-tagged POIs, photon counts were recorded from three points in the nucleus/chromatin and 2–3 points in the cytoplasm using the 488 nm laser with an exc. power between 0.6% and 1% depending on the protein expression level. Each FCS measurement lasted 30 s and only one repetition was performed per FCS point. To determine background fluorescence intensities and photon counts, FCS measurements of HK WT cells were performed. Similarly, in order to determine the count per molecule (CPM) of free EGFP, FCS measurements of HK WT cells expressing freely diffusing EGFP were conducted. These FCS measurements were taken manually at one point both in the nucleus/chromatin and the cytoplasm for 30 s each by using the same laser settings as described above for the EGFP-tagged POIs. An image indicating the FCS positions inside a cell was acquired before starting the FCS measurements. When data were acquired with the automatic workflow (30/34 data sets), a 3D image-stack was acquired at the end of the FCS-measurements for estimating the nuclear/chromatin volume.

## FCS analysis

Based on visual inspection of the images indicating the FCS positions, cells not corresponding to the desired cell cycle stage, dead cells and FCS points outside the cell or in the wrong subcellular compartment were excluded. From the remaining measurements, the autocorrelation function (ACF), a fit to a two-component diffusion model, and protein concentrations were determined (*Politi et al., 2018*). For several cell lines, the POI-EGFP concentration in the cytoplasm of interphase cells was so low that the recorded photon counts were close to background noise impairing a reliable estimation of the ACF. To account for this, only FCS measurements for which the fits fulfilled following conditions $R^2 > 0.92$, $\chi^2/N < 1.2$ ($N$ number of ACF time points) were included (~60% of the data). Further outliers were removed based on Tukey's fences of 3 times the interquartile range of CPMs and CPMs below 10 times the mean CPMs measured for mEGFP-C1. The last quality control left on average 59% of the total data.

The number of molecules listed in *Table 2* was estimated from the protein concentrations measured by FCS multiplied by the volumes of the respective compartments and the Avogadro's constant. For each cell, the nucleus/chromatin volume $V_{nuc}$ was obtained from the segmented chromatin signal of the 3D image-stack acquired at the end of the FCS measurement (*Walther et al., 2018*). In case of failed segmentation or lack of 3D images (<20% of the cases) the average volume for the specific cell-line and stage has been used. The cytoplasmic volume was calculated from $V_{cyt} = V_{cell} - V_{nuc} \approx V_{nuc}(V_r - 1)$ where we assume that the ratio of cell to nucleus/chromatin volume $V_r = V_{cell}/V_{nuc}$ is constant for a specific cell stage. The following ratios were used: G2/G1-phase $V_r = 3.04$; Prometaphase $V_r = 5.72$. The volume ratio values were obtained from 3D time-lapse imaging data of HK cell lines (*Cai et al., 2018*; *Walther et al., 2018*). These time-lapse imaging data sets include cellular volumes for mitotic stages from prophase, prometaphase up to cytokinesis from

over 600 cells. Prophase (mitotic standard phase 1) volumes were used as a proxy for G2 volumes. Prometaphase volumes (mitotic standard phase 5, 6, 7) were used for the volumes of nocodazole-arrested prometaphase cells.

Bootstrapping was used to compute statistics of the ratios between FCS measurements and data from LC-MS or from *Cai et al. (2018)* (*Appendix 1—table 4*, *Appendix 1—table 5*). For this, we sampled with replacement the measurements of two methods to compute 100,000 pairs. For each of these bootstrapped pair we compute the ratio. From the distribution of the ratios, we finally calculated the medians and 68% distribution intervals.

Whereas LC-MS measurements represent the average of a potentially diverse population of Hela cells, FCS-measurements rely on monoclonal populations of EGFP-tagged Hela cell lines. It is therefore possible that clonal variability (*Walther et al., 2018*) could contribute to the differences between our LC-MS and FCS measurements. FCS only measures fluorescent molecules, thus any protein in which the fluorophore is not visible, for example due to photobleaching or slow maturation, will not be detected. Also, FCS cannot detect completely immobile molecules, although it can detect local fluctuations of chromatin-bound molecules, which may arise either by movement of the chromatin itself or by diffusion of molecules along chromatin. We corrected for the presence of a non-diffusible pool by accounting for photobleaching (*Wachsmuth et al., 2015*; *Politi et al., 2018*), however it remains possible that our FCS measurements underestimated the number of chromatin-bound molecules.

## Mathematical modelling of cohesin dynamics in G1 and G2

The absolute copy number of nuclear cohesin $C_T$ is the algebraic sum of the copy number of unbound ($U$), dynamically-bound ($D$) and stably bound ($S$) cohesin forms:

$$C_T = U + D + S = U + B_T \tag{1}$$

where $B_T$ represents the sum of all chromatin-bound cohesin complexes. Assuming that the rates of chromatin binding and unbinding are balanced:

$$k_{on} \cdot U = k_{off} \cdot D \tag{2}$$

where the k's refer to the first order rate constants (with a dimension of time$^{-1}$) of individual steps in the mechanism. Using the definitions of $U = C_T - B_T$ and $D = B_T - S$:

$$k_{on}(C_T - B_T) = k_{off} \cdot (B_T - S) \tag{3}$$

the expression for $B_T$:

$$B_T = \frac{k_{on}}{k_{on} + k_{off}} C_T + \frac{k_{off}}{k_{on} + k_{off}} S \tag{4}$$

which is convenient to normalise to total cohesin level ($C_T$):

$$b_T = \frac{k_{on}}{k_{on} + k_{off}} + \frac{k_{off}}{k_{on} + k_{off}} s \tag{5}$$

by introducing the fraction of total ($b_T = B_T/C_T$) and stable ($s = S/C_T$) bound cohesin complexes. This equation tells us that stabilisation of cohesin on chromatin will lead to an increase of the chromatin-bound fraction. Furthermore, the increase of the chromatin-bound fraction ($b_T$) becomes a linear function of the stable fraction ($s$) if the kinetic parameters ($k_{on}$ and $k_{off}$) are constant, that is they are not influenced by the stabilisation of a fraction of cohesin on the chromatin. The linear relationship between the chromatin-bound ($b_T$) and stable ($s$) fractions has a slope of $\frac{k_{off}}{k_{on} + k_{off}}$ with an intercept of $\frac{k_{on}}{k_{on} + k_{off}}$. Notice that the sum of the slope and the intercept equals to one.

*Equation 5* allows us to plot the equilibrium distribution of different cohesin forms as a function of the stable fraction ($s$) (*Figure 4B*). The diagonal red line represents the fraction of the stable form ($s$) with a slope of one. Since $b_T = d + s$, the difference between the chromatin-bound ($b_T$) and stable ($s$) forms is the dynamic ($d$) form. The value of the unbound form ($u$) is given by the difference between the $b_T$ line (*Equation 5*) and one.

Cells in G1 phase do not have any stable cohesin ($s = 0$), therefore their bound-chromatin fraction ($b_T = 0.635$) defines the intercept of *Equation 5* (*Figure 4B,G1* black circle). Since the slope is (1 – intercept) which is equivalent to the fraction of unbound cohesin in G1 (0.365), *Equation 5* has the following parametric form:

$$b_T = 0.635 + 0.365 \cdot s \qquad (6)$$

The experimentally derived fraction of stable cohesin ($s$) in G2 phase cells is 0.367 (*Figure 3C*). According to *Equation 6*, at $s = 0.367$ the chromatin-bound fraction ($b_T$) should have a value of 0.769 (=0.635 + 0.365*0.367) (*Figure 4B,G2* black circle), consistent with the experimentally estimated value of 0.728 (*Figure 3C*).

## Acknowledgements

We thank the Vienna Biocenter Core Facilities for Illumina sequencing, M Julius Hossain for providing average cytoplasmic volumes of HeLa Kyoto cells at specific cell cycle stages, Malte Wachsmuth for discussion and Xavier Darzacq, Robert Tjian and Anders S Hansen for coordinating publication.

## Additional information

### Funding

| Funder | Grant reference number | Author |
|---|---|---|
| Boehringer Ingelheim | | Johann Holzmann<br>Kota Nagasaka<br>Johannes Fuchs<br>Gerhard Dürnberger<br>Wen Tang<br>Rene Ladurner<br>Georg A Busslinger<br>Karl Mechtler<br>Iain Finley Davidson<br>Jan-Michael Peters |
| Austrian Science Fund | FWF special research program SFB F34 | Jan-Michael Peters |
| Austrian Research Promotion Agency | FFG-834223 | Jan-Michael Peters |
| Vienna Science and Technology Fund | WWTF LS09-13 | Jan-Michael Peters |
| Seventh Framework Programme | 241548 (MitoSys) | Jan Ellenberg<br>Jan-Michael Peters |
| Horizon 2020 Framework Programme | 693949 | Jan-Michael Peters |
| Sixth Framework Programme | 503464 (MitoCheck) | Jan-Michael Peters<br>Jan Ellenberg |
| European Molecular Biology Organization | ALTF 1335-2016 | Kota Nagasaka |
| Human Frontier Science Program | LT001527/2017 | Kota Nagasaka |
| European Molecular Biology Laboratory | | Antonio Z Politi<br>Merle Hantsche-Grininger<br>Nike Walther<br>Birgit Koch<br>Jan Ellenberg |
| National Institutes of Health | Common Fund 4D Nucleome Program (U01 EB021223) | Jan Ellenberg |
| Paul G. Allen Frontiers Group | Allen Distinguished Investigator Program | Jan Ellenberg |

| | | |
|---|---|---|
| European Molecular Biology Laboratory | EMBL International PhD Programme (EIPP) | Nike Walther |
| Austrian Research Promotion Agency | FFG-852936 | Jan-Michael Peters |
| Austrian Research Promotion Agency | Laura Bassi Centre for Optimized Structural Studies grant FFG-840283 | Jan-Michael Peters |
| Austrian Science Fund | Wittgenstein award Z196-B20 | Jan-Michael Peters |
| Horizon 2020 Framework Programme | iNEXT 653706 | Jan Ellenberg |
| Horizon 2020 Framework Programme | 823839 | Karl Mechtler |
| Austrian Science Fund | I 3686-B25 MEIOREC - ERA-CAPS | Karl Mechtler |
| National Institutes of Health | Common Fund 4D Nucleome Program (U01 DA047728) | Jan Ellenberg |

The funders had no role in study design, data collection and interpretation, or the decision to submit the work for publication.

## Author contributions

Johann Holzmann, Conceptualization, Resources, Data curation, Formal analysis, Validation, Investigation, Visualization, Methodology, Writing—original draft, Writing—review and editing; Antonio Z Politi, Conceptualization, Resources, Data curation, Formal analysis, Validation, Investigation, Visualization, Methodology, Writing—review and editing; Kota Nagasaka, Conceptualization, Formal analysis, Validation, Investigation, Visualization, Writing—original draft, Writing—review and editing; Merle Hantsche-Grininger, Nike Walther, Formal analysis, Validation, Investigation, Writing—review and editing; Birgit Koch, Resources, Validation, Investigation; Johannes Fuchs, Formal analysis; Gerhard Dürnberger, Data curation, Formal analysis; Wen Tang, Rene Ladurner, Resources, Validation, Investigation, Visualization; Roman R Stocsits, Data curation, Formal analysis, Visualization; Georg A Busslinger, Validation, Investigation; Béla Novák, Conceptualization, Formal analysis, Funding acquisition; Karl Mechtler, Conceptualization, Supervision, Funding acquisition, Project administration; Iain Finley Davidson, Conceptualization, Formal analysis, Visualization, Writing—original draft, Project administration, Writing—review and editing; Jan Ellenberg, Conceptualization, Supervision, Funding acquisition, Project administration, Writing—review and editing; Jan-Michael Peters, Conceptualization, Supervision, Funding acquisition, Writing—original draft, Project administration, Writing—review and editing

## Author ORCIDs

Antonio Z Politi ![iD] https://orcid.org/0000-0003-4788-0933
Kota Nagasaka ![iD] http://orcid.org/0000-0003-0765-638X
Merle Hantsche-Grininger ![iD] https://orcid.org/0000-0002-5137-1616
Nike Walther ![iD] https://orcid.org/0000-0002-7591-5251
Béla Novák ![iD] http://orcid.org/0000-0002-6961-1366
Iain Finley Davidson ![iD] https://orcid.org/0000-0003-4945-6415
Jan Ellenberg ![iD] https://orcid.org/0000-0001-5909-701X
Jan-Michael Peters ![iD] https://orcid.org/0000-0003-2820-3195

## Decision letter and Author response

Decision letter https://doi.org/10.7554/eLife.46269.032
Author response https://doi.org/10.7554/eLife.46269.033

## Additional files

### Supplementary files
• Transparent reporting form
DOI: https://doi.org/10.7554/eLife.46269.018

### Data availability

Mass spectrometry proteomics data have been deposited to the ProteomeXchange Consortium via the PRIDE partner repository with the dataset identifier PXD012712. Sequencing data have been deposited in GEO (GSE126990).

The following datasets were generated:

| Author(s) | Year | Dataset title | Dataset URL | Database and Identifier |
|---|---|---|---|---|
| Holzmann J, Politi AZ, Nagasaka K, Hantsche-Grininger M, Walther N, Koch B, Fuchs J, Dürnberger G, Tang W, Ladurner R, Stocsits RR, Busslinger GA, Novak B, Mechtler K, Davidson IF, Ellenberg J, Peters J-M | 2019 | ChIP-seq data from Absolute quantification of cohesin, CTCF and their regulators in human cells | https://www.ncbi.nlm.nih.gov/geo/query/acc.cgi?acc=GSE126990 | NCBI Gene Expression Omnibus, GSE126990 |
| Holzmann J, Politi AZ, Nagasaka K, Hantsche-Grininger M, Walther N, Koch B, Fuchs J, Dürnberger G, Tang W, Ladurner R, Stocsits RR, Busslinger GA, Novak B, Mechtler K, Davidson IF, Ellenberg J, Peters J-M | 2019 | Mass spectrometry proteomics data | https://www.ebi.ac.uk/pride/archive/projects/PXD012712 | PRIDE Archive, PXD012712 |

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

## Appendix 1

DOI: https://doi.org/10.7554/eLife.46269.019

**Appendix 1—table 1. Peptide list.** Isotopically labelled reference peptides used to quantify cohesin subunit abundance and stoichiometry in HeLa cells. Note that SMC3 peptide ELAETEPK was replaced with peptide EQLSAER in one of the two SCC1 immunoprecipitation experiments.

| Protein | Peptide sequence |
|---------|------------------|
| SMC1 | VANYIK |
| SMC3 | ELAETEPK |
| SMC3 | EQLSAER |
| SMC3 | SNPYYIVK |
| SMC3 | TDLYAK |
| SMC3 | YYEVK |
| SCC1 | DVIDEPIIEEPSR |
| SCC1 | ETGGVEK |
| SCC1 | LIVDSVK |
| STAG1 | EDLLVLR |
| STAG1 | LELFTNR |
| STAG1 | YSADAEK |
| STAG2 | LELFTSR |
| STAG2 | YSVDAEK |

DOI: https://doi.org/10.7554/eLife.46269.020

**Appendix 1—table 2. LC-MS quantification of cohesin complex stoichiometry.** Quantification of cohesin subunits in SCC1 immunoprecipitates from chromatin or soluble extracts isolated from G1, G2 or prometaphase synchronised HeLa cells. Numbers are normalised relative to SCC1 abundance. Data are tabulated as mean [mean – s.d., mean + s.d.] from two biological replicates and two technical replicates. For individual peptide counts, see *Figure 1—figure supplement 4*.

| Protein | G1 chromatin-bound | Soluble | G2 chromatin-bound | Soluble | Prometaphase chromatin-bound | Soluble |
|---------|--------------------|---------|--------------------|---------|-------------------------------|---------|
| SMC1 | 0.89 [0.87, 0.91] | 0.86 [0.78, 0.94] | 0.9 [0.82, 0.98] | 0.77 [0.74, 0.8] | 0.88 [0.81, 0.95] | 0.84 [0.77, 0.91] |
| SMC3 | 0.95 [0.82, 1.08] | 0.96 [0.79, 1.13] | 0.97 [0.82, 1.12] | 0.99 [0.8, 1.18] | 0.98 [0.8, 1.16] | 0.98 [0.8, 1.16] |
| SCC1 | 1 [0.91, 1.09] | 1 [0.94, 1.06] | 1 [0.92, 1.08] | 1 [0.93, 1.07] | 1 [0.91, 1.09] | 1 [0.91, 1.09] |
| STAG1 | 0.15 [0.11, 0.19] | 0.04 [0.01, 0.07] | 0.11 [0.09, 0.13] | 0.04 [0.03, 0.05] | 0.13 [0.1, 0.16] | 0.08 [0.05, 0.11] |
| STAG2 | 0.68 [0.56, 0.8] | 0.7 [0.59, 0.81] | 0.74 [0.66, 0.82] | 0.76 [0.71, 0.81] | 0.75 [0.53, 0.97] | 0.73 [0.61, 0.85] |

DOI: https://doi.org/10.7554/eLife.46269.021

**Appendix 1—table 3. Protein concentration of cohesin subunits and regulators as measured by FCS.** Concentration of proteins in nM obtained from FCS measurements in the nucleus/chromatin (identified by the H2B-mCherry or SiR-DNA signal) and cytoplasmic compartment of cells. In G1 and G2 phase the protein concentrations in the cytoplasm were close to or below the detection limit, leading to >70% failed quality control (*Figure 2—source data 1*). Missing or italicised numbers indicate that the number of successful FCS measurements was not sufficient to estimate the protein concentration. Note that the EGFP-sororin cell line displayed a mitotic defect, raising the possibility that EGFP-sororin may be hypomorphic. Data are tabulated as the median; the 68% interval of the distribution is listed in brackets.

| Protein | G1 | | G2 | | Prometaphase | |
| | nucleus/chromatin | cytoplasm | nucleus/chromatin | cytoplasm | nucleus/chromatin | cytoplasm |
| --- | --- | --- | --- | --- | --- | --- |
| SCC1 | 332.00 [235.46; 464.19] | 5.65 [1.56; 14.57] | 290.03 [232.68; 494.03] | 2.92 [1.52; 4.73] | 98.55 [75.46; 135.37] | 83.40 [61.79; 109.13] |
| STAG1 | 70.53 [41.61; 128.47] | 1.32 [0.28; 3.79] | 85.12 [65.32; 141.73] | 0.45 [0.17; 1.53] | 24.15 [16.23; 35.94] | 15.95 [12.90; 21.11] |
| STAG2 | 282.54 [204.85; 375.89] | 11.58 [3.48; 37.06] | 281.55 [201.39; 372.99] | 14.14 [4.65; 44.77] | 105.42 [76.14; 149.19] | 85.54 [60.34; 116.91] |
| NIPBL | 196.95 [158.06; 300.70] | 4.74 [2.40; 9.11] | 166.18 [115.84; 223.47] | 4.58 [2.53; 9.64] | 56.83 [48.74; 75.62] | 59.53 [45.93; 69.90] |
| WAPL | 114.27 [86.52; 143.70] | - | 98.07 [80.84; 116.29] | - | 43.85 [35.51; 64.59] | 43.71 [37.76; 61.59] |
| SORORIN | 64.92 [35.49; 103.77] | 0.90 [0.30; 2.38] | 110.83 [53.43; 178.44] | 2.17 [0.44; 13.05] | 50.60 [35.37; 78.17] | 40.76 [29.19; 61.15] |
| CTCF | 187.65 [143.48; 247.89] | 2.16 [0.55; 3.36] | 166.60 [121.77; 226.64] | *3.65 [3.40; 3.91]* | 93.18 [60.86; 136.82] | 57.51 [34.12; 85.08] |

DOI: https://doi.org/10.7554/eLife.46269.022

**Appendix 1—table 4. Ratio of counts per molecule of EGFP-tagged proteins and monomeric mEGFP.** Distributions were estimated by bootstrapping the experimental measurements (100,000 repetitions with replacement). Data are tabulated as the median; the 68% interval of the distribution is listed in brackets.

| Protein | G1 | | G2 | | Prometaphase | |
| | nucleus/chromatin | cytoplasm | nucleus/chromatin | cytoplasm | nucleus/chromatin | cytoplasm |
| --- | --- | --- | --- | --- | --- | --- |
| SCC1 | 0.79 [0.58, 1.20] | 0.79 [0.58, 1.19] | 0.85 [0.60, 1.31] | 0.85 [0.60, 1.31] | 0.85 [0.63, 1.10] | 0.85 [0.63, 1.10] |
| STAG1 | 0.66 [0.39, 0.92] | 0.66 [0.39, 0.93] | 0.61 [0.19, 0.90] | 0.61 [0.19, 0.90] | 1.09 [0.81, 1.30] | 1.09 [0.80, 1.30] |
| STAG2 | 0.92 [0.60, 1.19] | 0.92 [0.60, 1.18] | 0.94 [0.56, 1.19] | 0.94 [0.56, 1.19] | 1.07 [0.75, 1.31] | 1.07 [0.75, 1.31] |
| NIPBL | 1.01 [0.74, 1.57] | 1.01 [0.74, 1.57] | 0.96 [0.72, 1.28] | 0.96 [0.72, 1.28] | 0.91 [0.65, 1.28] | 0.90 [0.65, 1.27] |
| WAPL | 1.15 [0.90, 1.51] | 1.15 [0.90, 1.51] | 1.21 [0.99, 1.57] | 1.21 [0.99, 1.57] | 0.87 [0.66, 1.06] | 0.87 [0.66, 1.06] |
| SORORIN | 1.14 [0.83, 1.62] | 1.14 [0.83, 1.61] | 1.07 [0.75, 1.67] | 1.07 [0.75, 1.67] | 0.67 [0.44, 0.88] | 0.67 [0.44, 0.89] |
| CTCF | 0.87 [0.69, 1.08] | 0.87 [0.69, 1.08] | 0.74 [0.57, 0.96] | 0.74 [0.57, 0.96] | 0.85 [0.65, 1.03] | 0.85 [0.65, 1.03] |

DOI: https://doi.org/10.7554/eLife.46269.023

**Appendix 1—table 5. Ratio of total protein copy number as estimated by FCS and LC-MS.**
Distributions were estimated by bootstrapping the experimental measurements (100,000 repetitions with replacement). Data are tabulated as the median; the 68% interval of the distribution is listed in brackets.

| | G1 | G2 | Prometaphase |
| --- | --- | --- | --- |
| **Protein** | **Ratio FCS:LC-MS** | **Ratio FCS:LC-MS** | **Ratio FCS:LC-MS** |
| SCC1 | 1.26 [0.79, 2.07] | 1.49 [0.89, 2.20] | 1.80 [1.21, 2.87] |
| STAG1 | 2.36 [1.25, 4.33] | 1.53 [0.93, 2.33] | 3.29 [2.37, 4.24] |
| STAG2 | 0.76 [0.55, 1.04] | 1.04 [0.76, 1.45] | 1.04 [0.77, 1.37] |

DOI: https://doi.org/10.7554/eLife.46269.024

**Appendix 1—table 6. Number of peaks called following SMC3, STAG1, STAG2 and CTCF ChIP-Seq in G1 phase.** Peaks called using MACS peak caller version 1.4.2 with a *p*-value threshold of 1e-10.

| Protein | Number of ChIP-seq peaks |
| --- | --- |
| SMC3 | 36713 |
| STAG1 | 35068 |
| STAG2 | 47063 |
| CTCF | 41502 |

DOI: https://doi.org/10.7554/eLife.46269.025

**Appendix 1—table 7. Counts of solitary and co-localising peaks called following SMC3, STAG1, STAG2 and CTCF ChIP-Seq in G1 phase.** SMC3, CTCF, and the combined set of STAG1 and STAG2 Chip-seq peaks identified in G1 HeLa cells. All subsets are shown.

| Overlap | Number of ChIP-seq peaks |
| --- | --- |
| SMC3 alone | 1895 |
| STAG1/2 alone | 13718 |
| CTCF alone | 5842 |
| SMC3 + STAG1/2 | 4234 |
| SMC3 + CTCF | 2286 |
| STAG1/2 + CTCF | 5402 |
| SMC3 + STAG1/2 + CTCF | 27725 |

DOI: https://doi.org/10.7554/eLife.46269.026

