## [Decision Letter]

Thank you for submitting your article "Absolute quantification of cohesin, CTCF and their regulators in human cells" for consideration by *eLife*. Your article has been reviewed by three peer reviewers, including David J Sherratt as the Reviewing Editor and Reviewer #1, and the evaluation has been overseen by Kevin Struhl as the Senior Editor.

The reviewers have discussed the reviews with one another and the Reviewing Editor has drafted this decision to help you prepare a revised submission.

We are unanimous in recommending publication of your paper (along with the complementary paper of Cattoglio and colleagues) after minor revisions that attend to the points below are addressed.

The manuscript focuses on the absolute quantification of protein copy numbers of cohesin complex components in human cells. This information is fundamental in the genome organization field, as it allows to set some constraints to models that have been previously proposed for the role of this complex in determining chromosome folding in topologically associated domains and in chromatin loops. The copy-number of cohesin subunits measured by the authors at different cell cycle stages, using two orthogonal approaches, mass spectrometry and fluorescence correlation spectroscopy are in reasonable agreement with each other. Similarly, to what is proposed in Cattoglio et al., the conclusions of the authors point to a model in which not all of the possible cohesin binding sites in the mammalian genome are simultaneously occupied in a single cell. A significant implication of these numbers is that cohesin is likely to frequently pass CTCF sites when these are unoccupied by CTCF.

Essential revisions:

1) Differences in copy number estimates between LC-MS and FCS need to be discussed more fully. If ones looks at the total copy number of cohesin measured by the two techniques (that is chromatin bound + soluble for MS, and nuclear + cytosolic for FCS), FCS seems to provide a lower estimate for all of the tested conditions (except one: STAG2 in G1). In some cases the FCS estimate is around half the MS estimate. I think that the authors should try to explain this potential discrepancy. One potential bias in FCS is related to the potential homo-oligomerization state of cohesin subunits while they diffuse in the nucleus of the cell. This is due to the fact that the G(0) of the FCS curve measures the concentration as the ratio of the average intensity in the observation volume divided by the fluctuation in fluorescence brought by one diffuser moving in and out of the focal volume. If some of the diffusers are dimeric (and carry two fluorescent tags), this might lead to an underestimation of the monomer concentration. I am not expert in cohesin, but I understand that the possibility of cohesin subunits to homo-dimerize in mammals is debated. However, it would be nice if the authors could exclude that their FCS data captures some homo-oligomers. This could probably be done by analyzing the first- and second- moment of their fluorescence fluctuations, as it is done in the Number and Brightness (N&B) approach (Digman et al., Biophys J, 2008). Also, it would be helpful if the authors could comment on the impact of a large fraction of stably bound molecules on the concentration estimates by FCS. My understanding is that both mobile and immobile molecules would contribute to the measured concentration, but this needs to be explicitly mentioned in the text: it is a little counterintuitive how FCS can provide information about total concentration (immobile + mobile), since the technique relies on measuring the fluctuations of intensity brought by mobile molecules only.

2) From the data in Figure 1—figure supplement 4, it is difficult to conclude that the stoichiometry of cohesin subunits is different in the different phases of the cell cycle. At minimum the authors should try to test statistically (within a certain interval of confidence) that their data is incompatible with the stoichiometry 1:1:1:0.75 in G2. At the moment the data seems to be compatible with both the monomeric ring model and with the handcuffs model.

3) The "Modeling" section. The scheme of the kinetic reaction in Figure 4A seems to imply that the stable binding of cohesion occurs through an intermediate dynamic binding event. However, another possibility is that the stable binding events are independent (and competing) with the dynamic events, therefore controlled by a second set of kinetic rates. Equation 2 should still be valid in this case (as it can be derived from the steady state of the D population), as well as all the other equations, so their theory should hold also in the case of competitive binding between stable and dynamic events. Nevertheless, the authors could mention this possibility. Also, the main text describing this part is not very clear. The authors mention that the mathematical modelling is performed "to test whether an increase in cohesin's residence time is sufficient to explain the change in the number of chromatin-bound cohesin complexes, or other alterations in cohesin dynamics might also contribute", but the authors do not mention what these other alterations refer to. To one reviewer, the mathematical modelling simply determines that the observed increase in bound fraction in G2 is entirely due to the appearance of a more stable binding state on DNA in this phase. Finally, the scheme in Figure 4B is not explained with sufficient detail. I assume that the red line is s, and the grey line is b_t_, that the upper black dots are the total bound fraction measured in G1 and G2, while the bottom black dots are the measurements for s in G1 and G2, but a more detailed description should be added to the figure legend.

4) In Figure 3 and Figure 3—figure supplement 1 the authors show FRAP experiments for SCC1. The example shown for G1 is not very convincing. It seems that the lower part of the nucleus is not fully bleached, and also that this cell is a bit out of focus. Figure 3—figure supplement 1G shows the reduction in signal in the unbleached area between pre-bleach and post-bleach. A difference could also occur due to non-specific photobleaching of the whole frame. It would therefore be good to include the difference in intensity pre- and post-bleach for a control cell from the same frame.

5) Could the authors explain why SMC1 is not quantified in the soluble fraction of the LC-MS experiments? Is this due to the use of only one reference peptide for SMC1?

6) The authors measure the stoichiometry of cohesin complexes by pulling on one subunit and measuring the abundance of the other subunits. As the composition of the complex is 1:1:1:1 in G2, they conclude that the 'handcuff model' in which one STAG subunit bridges two complexes after DNA replication is unlikely to be correct. I agree, but in all fairness this particular handcuff model made no sense from the start anyway. These results however do not rule out the existence of cohesin dimers of full complexes. Such a 2:2:2:2 stoichiometry is hard to distinguish from 1:1:1:1, which could be mentioned in the text. This is of particular interest as cohesin dimers have been proposed to play a role in loop formation, and also as the co-submitted Cattoglio paper does indeed observe dimers/multimers.

7) Supplementary Figure 7 shows the ChIP data for cohesin subunits and CTCF. It could be nice to move this figure to the main figures and to then include a schematic depiction of the implications for the abundance of cohesin and CTCF on chromatin in single cells.

8) The two slightly different green colours in Figure 2 are noteasy to distinguish for the reader. Perhaps one of these colours could be changed into something more distinct into something more distinct.

9) One reviewer thought that a final schematic summarizing the results in cartoons representing cells in G1/S etc. with some sort of simply visual representation of the relative numbers/classes of the analyzed molecules would be helpful to the general reader.

---

## [Author Response]

Essential revisions:1) Differences in copy number estimates between LC-MS and FCS need to be discussed more fully. If ones looks at the total copy number of cohesin measured by the two techniques (that is chromatin bound + soluble for MS, and nuclear + cytosolic for FCS), FCS seems to provide a lower estimate for all of the tested conditions (except one: STAG2 in G1). In some cases the FCS estimate is around half the MS estimate. I think that the authors should try to explain this potential discrepancy.

To address this issue we computed additional statistics and highlighted possible reasons for the discrepancy between our LC-MS and FCS datasets in the Discussion and Materials and methods sections. We also added Table 7, which directly compares our LC-MS and FCS measurements of absolute copy number.

One potential bias in FCS is related to the potential homo-oligomerization state of cohesin subunits while they diffuse in the nucleus of the cell. This is due to the fact that the G(0) of the FCS curve measures the concentration as the ratio of the average intensity in the observation volume divided by the fluctuation in fluorescence brought by one diffuser moving in and out of the focal volume. If some of the diffusers are dimeric (and carry two fluorescent tags), this might lead to an underestimation of the monomer concentration. I am not expert in cohesin, but I understand that the possibility of cohesin subunits to homo-dimerize in mammals is debated. However, it would be nice if the authors could exclude that their FCS data captures some homo-oligomers. This could probably be done by analyzing the first- and second- moment of their fluorescence fluctuations, as it is done in the Number and Brightness (N&B) approach (Digman et al., Biophys J, 2008).

We thank the reviewers for their useful comments. Unfortunately, we can’t apply the N&B method to our current data since N&B requires videos acquired at a short time-interval. Instead, we compared the photon counts per molecule of our proteins of interest (CPM_POI) to the values obtained from cells expressing monomeric EGFP only (CPM_mEGFP). One would expect that the CPM_POI is higher for homo-oligomers than for CPM_mEGFP. For all proteins, the ratio r_CPM = CPM_POI/ CPM_mEGFP was close to 1 and ranged from 0.7 – 1.18 with a standard deviation of around 0.3 (new Table 5). For 3 out of 7 proteins we observed in only one of the replicates a small but significant increase in CPM_POI compared to CPM_mEGFP (Wilcoxon test p < 0.01). Although this does not completely rule out that some homo-oligomers may be present, our data suggests that this fraction is small compared to the overall protein fraction. We included a paragraph at the beginning of the section ‘Fluorescence correlation spectroscopy analysis of cohesin, CTCF and other cohesin regulators’ to discuss this point.

Also, it would be helpful if the authors could comment on the impact of a large fraction of stably bound molecules on the concentration estimates by FCS. My understanding is that both mobile and immobile molecules would contribute to the measured concentration, but this needs to be explicitly mentioned in the text: it is a little counterintuitive how FCS can provide information about total concentration (immobile + mobile), since the technique relies on measuring the fluctuations of intensity brought by mobile molecules only.

It is correct that FCS only accounts for diffusing molecules, however, this also includes local fluctuations of chromatin-bound molecules. These fluctuations may arise either by movement of the chromatin itself or by diffusion of cohesin on chromatin, which has been predicted to occur based on a number of experimental observations (see Discussion section ‘Occupancy of cohesin and CTCF enrichment sites in a single cell’ for details). To compute the concentration in the nucleus, we account for an immobile fraction by correcting for bleaching as described in Wachsmuth et al., 2015. We have also compared our average STAG1, STAG2, WAPL, CTCF and SCC1 concentrations with previously published data using the same cell lines (Cai et al., 2018). In Cai et al., 2018 we used FCS-calibrated imaging to estimate concentrations in non-synchronized dividing cells. For FCS-calibrated imaging, one uses image fluorescence ratios, which do not depend on an immobile fraction, and a calibration curve derived from FCS-concentration measurements of cells expressing free diffusing monomeric GFP (mEGFP). We found that the concentrations in the nucleus/chromatin compartments of cells synchronized in prometaphase were very similar to those reported in Cai et al., 2018 for mitotic cells in prometaphase (ratio 1.00 +- 0.54). The same is true for the cytoplasm of prometaphase cells (ratio 1.00 +- 0.52). Since the only cell cycle stage common to both datasets is prometaphase, during which little cohesin or cohesin regulators are chromatin-bound, we decided not to include this comparison in our manuscript. However we expanded on the potential reasons for the discrepancy between our LC-MS and FCS datasets in the Discussion and Materials and methods sections.

2) From the data in Figure 1—figure supplement 4, it is difficult to conclude that the stoichiometry of cohesin subunits is different in the different phases of the cell cycle. At minimum the authors should try to test statistically (within a certain interval of confidence) that their data is incompatible with the stoichiometry 1:1:1:0.75 in G2. At the moment the data seems to be compatible with both the monomeric ring model and with the handcuffs model.

We have modified this section to describe why we think these experiments underestimate the stoichiometry of SMC1, SMC3 and STAG1/2 relative to SCC1. We have also calculated the 95% confidence interval of the ratio of STAG1/2 to SCC1 in G2 (0.77 – 0.93). Since this interval is greater than 0.75, and is likely to be lower than the true STAG1/2:SCC1 ratio, we think our data is compatible with the monomeric ring model. We have included this confidence interval and our reasoning in the text.

3) The "Modeling" section. The scheme of the kinetic reaction in Figure 4A seems to imply that the stable binding of cohesion occurs through an intermediate dynamic binding event. However, another possibility is that the stable binding events are independent (and competing) with the dynamic events, therefore controlled by a second set of kinetic rates. Equation 2 should still be valid in this case (as it can be derived from the steady state of the D population), as well as all the other equations, so their theory should hold also in the case of competitive binding between stable and dynamic events. Nevertheless, the authors could mention this possibility.

Thank you for noting this interesting possibility, however we feel that this is unlikely to be the case. If the stable binding events occur independently from the dynamic binding events and stable binding and unbinding is in equilibrium, then:

kon'∙U=koff'∙S

where the k’s represent the rate constants of this reaction, U = soluble cohesin and S = stably chromatin-bound cohesin. These rate constants are ln^2^/(half-life). Stably chromatin-bound cohesin has a half-life of around 10 hours; in G2, U = 0.27 and S = 0.37. Using the above equation, this would mean that the half-life for the on-rate would be 13.5 hours; i.e. it would take 13.5 hours for stably chromatin-bound cohesin to reach 50% of the steady state level if it binds independently of the dynamic population. This does not fit with the observed kinetics of the appearance of stably chromatin-bound cohesin and is not compatible with cohesin’s ability to generate and maintain sister chromatid cohesion.

Also, the main text describing this part is not very clear. The authors mention that the mathematical modelling is performed "to test whether an increase in cohesin's residence time is sufficient to explain the change in the number of chromatin-bound cohesin complexes, or other alterations in cohesin dynamics might also contribute", but the authors do not mention what these other alterations refer to. To one reviewer, the mathematical modelling simply determines that the observed increase in bound fraction in G2 is entirely due to the appearance of a more stable binding state on DNA in this phase.

We agree that the mathematical modeling indicates that the observed increase in chromatin-bound cohesin is due to the appearance of stably-bound cohesin. We have modified the following sentence:

‘To test whether an increase in cohesin's residence time is sufficient to explain these changes, or whether other alterations in cohesin dynamics might also contribute, we performed mathematical modeling.’

To this:

‘To explore whether the observed increase in cohesin's residence time is sufficient to explain the two-fold increase in chromatin-bound cohesin complexes, we performed mathematical modeling.’

Finally, the scheme in Figure 4B is not explained with sufficient detail. I assume that the red line is s, and the grey line is b_t_, that the upper black dots are the total bound fraction measured in G1 and G2, while the bottom black dots are the measurements for s in G1 and G2, but a more detailed description should be added to the figure legend.

We have added these details into the legend for Figure 4.

4) In Figure 3 and Figure 3—figure supplement 1 the authors show FRAP experiments for SCC1. The example shown for G1 is not very convincing. It seems that the lower part of the nucleus is not fully bleached, and also that this cell is a bit out of focus. Figure 3—figure supplement 1G shows the reduction in signal in the unbleached area between pre-bleach and post-bleach. A difference could also occur due to non-specific photobleaching of the whole frame. It would therefore be good to include the difference in intensity pre- and post-bleach for a control cell from the same frame.

We have replaced the upper panels in Figure 3A and Figure 3—figure supplement 1A with better examples of cells that were properly focused during iFRAP. We agree that it is important to determine whether non-specific photobleaching contributed to the reduction in signal between pre-bleach and post-bleach. We measured the difference in fluorescence intensity within the nuclei of 17 control cells within the same fields of view used to acquire fluorescence intensity changes in cells following iFRAP. There was no significant decrease in signal in these control cells. We included this data in Figure 3—figure supplement 1G.

5) Could the authors explain why SMC1 is not quantified in the soluble fraction of the LC-MS experiments? Is this due to the use of only one reference peptide for SMC1?

Our study relied on a single reference peptide for SMC1, which unfortunately was filtered out from the soluble LC-MS measurements during quantification. We have now described this in the Materials and methods section.

6) The authors measure the stoichiometry of cohesin complexes by pulling on one subunit and measuring the abundance of the other subunits. As the composition of the complex is 1:1:1:1 in G2, they conclude that the 'handcuff model' in which one STAG subunit bridges two complexes after DNA replication is unlikely to be correct. I agree, but in all fairness this particular handcuff model made no sense from the start anyway. These results however do not rule out the existence of cohesin dimers of full complexes. Such a 2:2:2:2 stoichiometry is hard to distinguish from 1:1:1:1, which could be mentioned in the text. This is of particular interest as cohesin dimers have been proposed to play a role in loop formation, and also as the co-submitted Cattoglio paper does indeed observe dimers/multimers.

We agree that it is important to draw attention to the caveat that our approach is unable to distinguish between 1:1:1:1 and 2:2:2:2. We have added a sentence to this effect in the Results subsection “The stoichiometry of cohesin complexes remains constant throughout G1, G2 and prometaphase”.

7) Supplementary Figure 7 shows the ChIP data for cohesin subunits and CTCF. It could be nice to move this figure to the main figures and to then include a schematic depiction of the implications for the abundance of cohesin and CTCF on chromatin in single cells.

We have prepared a new main figure (Figure 5), which includes the ChIP-seq data from Supplementary Figure 7 and a schematic depiction of the implications for the abundance of cohesin and CTCF on chromatin loop formation in single cells.

8) The two slightly different green colours in Figure 2 are noteasy to distinguish for the reader. Perhaps one of these colours could be changed into something more distinct into something more distinct.

We have changed the colour scheme in Figure 2 and throughout the manuscript.

9) One reviewer thought that a final schematic summarizing the results in cartoons representing cells in G1/S etc. with some sort of simply visual representation of the relative numbers/classes of the analyzed molecules would be helpful to the general reader.

Unfortunately, we could not find a way to summarise our data regarding the relative numbers and classes of analysed molecules. We hope that our newly added schematic (Figure 5D) provides a good overview of the key findings of our study.